# Online Combustion Status Recognition of Municipal Solid Waste Incineration Process Using DFC Based on Convolutional Multi-Layer Feature Fusion

**Xiaotong Pan** [1,2]**, Jian Tang** [1,2,*]**, Heng Xia** [1,2] **and Tianzheng Wang** [1,2]

1 Faculty of Information Technology, Beijing University of Technology, Beijing 100124, China; pxt@emails.bjut.edu.cn (X.P.); xiaheng@emails.bjut.edu.cn (H.X.); wangtz@emails.bjut.edu.cn (T.W.)
2 Beijing Laboratory of Smart Environmental Protection, Beijing 100124, China
* Correspondence: freeflytang@bjut.edu.cn or freeflytang@126.com

**Abstract:** The prevailing method for handling municipal solid waste (MSW) is incineration, a critical process that demands safe, stable, and eco-conscious operation. While grate-typed furnaces offer operational flexibility, they often generate pollution during unstable operating conditions. Moreover, fluctuations in the physical and chemical characteristics of MSW contribute to variable combustion statuses, accelerating internal furnace wear and ash accumulation. Tackling the challenges of pollution, wear, and efficiency in the MSW incineration (MSWI) process necessitates the automatic online recognition of combustion status. This article introduces a novel online recognition method using deep forest classification (DFC) based on convolutional multi-layer feature fusion. The method entails several key steps: initial collection and analysis of flame image modeling data and construction of an offline model utilizing LeNet-5 and DFC. Here, LeNet-5 trains to extract deep features from flame images, while an adaptive selection fusion method on multi-layer features selects the most effective fused deep features. Subsequently, these fused deep features feed into DFC, constructing an offline recognition model for identifying combustion status. Finally, embedding this recognition system into an existing MSWI process data monitoring system enables online flame video recognition. Experimental results show remarkable accuracies: 93.80% and 95.08% for left and right grate furnace offline samples, respectively. When implemented in an online flame video recognition platform, it aptly meets recognition demands.

**Keywords:** municipal solid waste incineration; combustion status; LeNet-5 network; deep forest classification; online flame video identification

## 1. Introduction

Municipal solid waste incineration (MSWI) serves as a sustainable approach method for effectively managing the challenges posed by municipal solid waste (MSW) in terms of environmental sustainability [1]. Through high-temperature combustion, it transforms MSW into ash and heat energy, playing a pivotal role in tackling the escalating environmental issues associated with MSW treatment [2]. It also mitigates, to a certain extent, the negative impacts of conventional landfilling and composting practices on the environment. However, with the increasing emphasis on environmental sustainability, there is a heightened focus on the feasibility and long-term consequences of MSWI. Potential challenges arise in the MSWI process, including the release of harmful gases that can compromise air and water quality, posing a substantial threat to environmental sustainability [3]. Consequently, it is imperative to implement effective control measures in the design and operation of incineration facilities to reduce emissions and minimize their impact on the environment and human health. In the pursuit of sustainability, MSWI must strike a delicate balance encompassing economic, social, and environmental considerations.

MSWI has gained widespread global recognition owing to its substantial benefits in terms of harm reduction, minimization, and resource utilization [4,5]. A variety of incinerators exist for MSW, such as grate-type, bed-type, and fluidized bed incinerators. The predominant method for the MSWI process typically employs grate-type incinerators. [6]. Compared to other furnace types, grate-based ones are known for their characteristics of flexibility and easy operation. However, their energy efficiency is low, and their pollutant-emission rate is high under unstable running status [7]. Technological innovation emerges as a key element in achieving this equilibrium, fostering the green and standardized management of MSW through techniques [8,9]. This will propel MSW management toward a more sustainable trajectory. Thus, much more advanced technologies, such as machining learning and artificial intelligence based on vision, are needed to overcome these problems [10]. Due to its high heterogeneity, MSW poses challenges in maintaining combustion stability, potentially resulting in issues such as coking, ash accumulation, and corrosion inside the furnace. Therefore, making timely and accurate judgments on combustion status becomes necessary [11].

Presently, the observation of waste incinerator combustion status primarily relies on visual assessments by experts. They combine visual observations with flame conditions from on-site observation holes to adjust key parameters, ensuring combustion stability [12]. However, this method faces several challenges: (1) the lack a unified judgment standard leads to inconsistent results, susceptible to subjective variations; (2) prolonged on-site image observation induces visual fatigue in workers, impacting their health; (3) multiple interrelated key regulatory parameters significantly impact combustion efficiency, making accurate individual control by operators extremely challenging, potentially causing unstable control processes. Relying solely on manual methods for identifying combustion status is no longer adequate to meet production requirements. To enhance on-site detection automation, reduce subjective influences stemming from human factors, decrease labor intensity, and improve detection efficiency, employing online flame video recognition technology based on artificial intelligence is crucial [13].

When it comes to recognizing combustion status through flame-image analysis in the MSWI process, several studies exist, each focusing on different furnace types. Miyamoto et al. [14] conducted research on the "AI-VISION" system, integrating combustion-image processing, neural networks for discerning combustion status, and online learning methods for optimizing neural networks. Their system manipulated operating values in fluidized bed incinerators. Zhou [15] developed a combustion status diagnosis model based on neural networks utilizing geometric features and grayscale information from flame images, validated through ten-fold cross-validation experiments. Guo et al. [16] presented a combustion status-recognition method employing mixed data augmentation and a deep convolutional generative adversarial network (DCGAN) to obtain flame images under diverse conditions. Huang et al. [12] extracted key parameters like grayscale mean, flame area ratio, high-temperature ratio, and flame front to characterize and evaluate combustion status. Meanwhile, Zhang et al. [17] extracted 19 feature vectors encompassing color, shape, and texture of flame images, constructing an echo state network recognition model. These findings emphasize the necessity for further research and validation of combustion status identification methods tailored to different MSWI plants. In the field of combustion status recognition based on flame videos, researchers have proposed diverse solutions for similarly complex industrial processes. Chen et al. [18] utilized typical video blocks of rotary kiln flame combustion as model training samples. They extracted texture and motion features from these blocks and inputted them into a support vector machine (SVM) to construct a flame status-recognition model, though with relatively unstable recognition performance. Li et al. [19] employed a convolutional recurrent neural network (CRNN) with spatiotemporal relationships from rotary kiln flame image sequences to predict combustion status. Wu et al. [20] initially used a convolutional neural network (CNN) to extract spatial features from electric melting magnesia furnace video signals. Then, they applied a recurrent neural network (RNN) to extract temporal features, achieving automatic labeling

of abnormal conditions using weighted median filtering. These studies indicated that flame video recognition is founded upon analyzing sequences of flame images. Thus, achieving video recognition of combustion status in the MSWI process should commence with constructing an offline recognition model based on flame images.

The offline modeling process for flame-image recognition typically comprises two stages: feature extraction and image recognition. Some researchers have focused on manual feature extraction methods to derive flame features. For instance, Zhang et al. [21] extracted multiple feature vectors encompassing color, shape, and texture features from flame images, utilizing these as inputs to the bilinear convolutional neural network (BCNN) for flame-image recognition. Wu et al. [22] initially segmented the pertinent region in the flame image and, subsequently, employed extracted color, texture, and rectangularity features for flame recognition. Another approach by Wu et al. [23] assessed image quality by modeling texture, structure, and naturalness, using the resulting image quality score as the input for the visual recognition model. However, the ability of the extracted feature parameters in the aforementioned studies to accurately represent combustion status relies partly on image-processing techniques, such as image segmentation algorithms, and partly on manual expertise. Consequently, this approach has significant limitations and inherent instability.

Feature extraction methods based on deep learning offer the capability to autonomously learn representative features from flame images. Han et al. [24] utilized flame images to train the convolutional sparse autoencoder (CSAE), resulting in a feature extractor adept at extracting deep features. Visualization of these features demonstrated clear discriminability across various combustion statuses. Similarly, Liu et al. [25] applied deep learning to industrial combustion processes, employing a multi-layered deep belief network (DBN) to extract nonlinear features. This approach yielded descriptive insights into flame physical properties, outperforming traditional principal component analysis (PCA). These studies validated the immense potential of deep networks in combustion status recognition. LeNet-5, a convolutional neural network devised by LeCun et al. in 1998, gained prominence in handwritten digit recognition, showcasing commendable recognition results [26]. Roy et al. [27] utilized LeNet-5 to extract deep features from forest fire images, offering insights for developing early-stage forest fire detection systems by controlling model complexity through L2 regularization. He et al. [28] enhanced the model by augmenting the layer count of the LeNet-5 network and incorporating a dropout layer, achieving heightened recognition accuracy. Li et al. [29] merged low-level and high-level features extracted from the LeNet-5 structure, leveraging the first two pooling layers and fully connected layers as SoftMax inputs for micro expression recognition, yielding robust results on a public expression database. LeNet-5's capability to capture local image features based on local receptive fields, reduce network training parameters through shared weights, and maintain a simple network structure is noteworthy. Despite being an early convolutional neural network with shallow layers, LeNet-5 finds extensive use in image-processing tasks like license plate recognition and face detection. These studies show LeNet-5's broad application prospects in image recognition. Its characteristic structure excels in extracting deep features, making it a promising choice for MSWI flame combustion status recognition in this study.

Drawing inspiration from deep neural network models, the deep forest classification (DFC) algorithm introduced by Zhou et al. [30] comprises two primary components: a multi-grained scan and a cascaded forest (CF). The former transforms raw data features, while the latter constructs prediction models using these transformed features [31,32]. The multi-grained scan bolsters CF training, augmenting its effectiveness. Cao et al. [33] integrated a rotating forest into the cascaded layer to enhance DFC's discriminative ability for hyperspectral features. Their work also leveraged spatial information from adjacent pixels, refining hyperspectral image classification. Zheng et al. [34] tackled challenges in leaf classification, specifically addressing the lack of large-scale professional datasets and expert knowledge annotations. They utilized generative adversarial networks for image feature extraction and a designed fuzzy random forest as CF's base learner, achieving

superior recognition performance compared to existing techniques. Sun et al. [35] applied DFC to chest computer tomography (CT) scan image recognition for coronavirus disease-19 (COVID-19). Extracting features from specific image locations, they employed DFC to learn high-level representations, resulting in commendable recognition performance. Additionally, Nie et al. [36] proposed an online multi-view deep forest architecture for remote sensing image data. DFCs offer advantages over DNNs, such as fewer hyperparameters, interpretability, and automatic adjustment of model complexity [37]. Moreover, they perform well with smaller image data samples, effectively resolving challenges in constructing DNN recognition models. However, it is noteworthy that the multi-grained scan module of DFC can be time-consuming and inefficient in acquiring diverse scaled deep features. These studies collectively imply that DFC, combined with CNN-based deep feature extraction algorithms, can effectively tackle the limitations posed by limited flame-image datasets in the MSWI process.

In summary, achieving online recognition of combustion video status in the MSWI process entails addressing several key factors: (1) effectively extracting deep features from flame images despite limited sample size; (2) maximizing the utilization of these extracted deep features to build a recognition model that meets on-site recognition requirements; (3) advancing toward online video recognition by leveraging flame-image recognition. Hence, this article proposes an online video recognition method rooted in convolutional multi-layer feature fusion and DFC. This method involves (1) training the LeNet-5 network using flame images collected on-site to extract deep flame features; (2) employing an adaptive fusion method based on LeNet-5 multi-layer features to select and use fused features as flame representations; (3) utilizing the extracted deep fusion features in DFC to construct an offline recognition model for determining combustion status based on flame images; and (4) integrating the offline recognition algorithm into the developed MSWI flame video combustion status-recognition platform to achieve real-time online recognition.

The existing research highlights prevalent applications of online flame video recognition in areas like rotary kilns and electric magnesium melting furnaces. Surprisingly, there is a dearth of studies regarding online flame video recognition in the MSWI field. Consequently, this article aims to explore an online recognition method tailored to the unique characteristics of flame videos in MSWI. The primary innovations of this method encompass (1) proposing a fusion technique that combines flame depth feature extraction and adaptive selection based on LeNet-5; (2) integrating deep fusion features with the DFC algorithm to construct a combustion status-recognition model specifically designed for the MSWI process; and (3) developing a practical online combustion status-recognition platform based on flame video for MSWI. These advancements signify the potential practical value of this technology within the MSWI field.

## 2. Flame-Image Analysis of the MSWI Process for Online Recognition

### 2.1. Description of Flame Image in the Furnace

Figure 1 shows the process flow of grate-type MSWI in Beijing.

The MSWI process includes six stages: solid waste storage and transportation, solid waste combustion, use of a heat recovery boiler, steam electric power generation, flue gas cleaning, and flue gas emission. Initially, MSWs undergo collection and transportation via vehicles to the MSWI plant, where they undergo fermentation and dehydration to attain a high calorific value. Subsequently, these wastes are elevated and deposited into the feed hopper of the incinerator. Within this phase, the feeder pushes MSWs into the incinerator, traversing through various stages: drying, burning 1, burning 2, and burnout. The flue gas generated by combustion is then directed by the induced draft fan into the waste heat recovery system, generating high-temperature steam through heat exchange with liquid water in the boiler drum. Exiting the boiler outlet, the flue gas proceeds successively through the reactor and bag filter. Ultimately, the induced draft fan discharges the flue gas from the stack into the environment after the removal of acidic gases, particles, and active

carbon adsorbates. This emission phase marks the presence of components such as HCl, $SO_2$, NOx, dioxins, and other substances [38].

**Figure 1.** Process flow of an MSWI plant in Beijing.

From the solid waste combustion stages depicted in Figure 1, industrial cameras are positioned at oblique upper positions on the end of grates to capture real-time flame video streams. These videos are then transmitted via coaxial cables to the supervisory control room of the distributed control system (DCS). In this study, video acquisition cards are utilized to store these streams on the data acquisition computer for offline modeling. Typically, field experts assess the combustion status of municipal solid waste (MSW), and the corresponding manipulation strategy controls the MSWI process. Consequently, combustion status serves as key feedback information for achieving intelligent control of the MSWI process.

### 2.2. Combustion Status Analysis of MSWI Process

Figure 2 illustrates the correspondence between the layout of the furnace grate within the incinerator and the captured flame image.

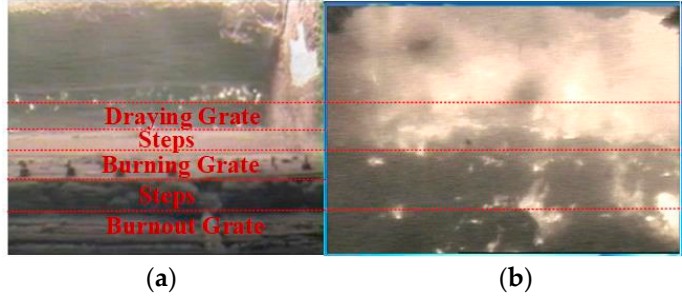

(**a**)　　　　　　(**b**)

**Figure 2.** Correspondence relation between furnace grate and flame-image distribution: (**a**) furnace grate image; (**b**) flame image.

In Figure 2a, the interior screen of the right-side furnace clearly displays the layout, featuring the dry grate, combustion grates 1 and 2, the burning grate, and the steps between these grates. This provides a clear means to determine the flame-burning position by

aligning the furnace-grate image. Before making the classification of combustion status, our preliminary investigation focused on abnormal combustion phenomena within biomass grate furnace combustion. Huang [15] defined layered combustion deviation status while studying diagnostic methods for the MSWI process, highlighting lateral and longitudinal deviations in the flame's spatial distribution. In the field of biomass grate furnace combustion, Duffy [39] and other researchers [40] identified a phenomenon termed "channeling." This occurs when the bed inside the combustion chamber is uneven or at the junction with the furnace's boundary wall. Channeling disrupts the uniformity of the secondary air blown in from beneath the grate, exacerbating bed irregularities. This article classifies MSWI flame images into four distinct combustion statuses: normal, deviation, channeling, and smoldering. This classification draws from the observed on-site flame combustion conditions in the studied MSWI process and the analysis of abnormal disturbance phenomena in grate furnace combustion, using knowledge from on-site experts and research scholars. Following the effective classification, corresponding adjustments to control strategies will be initiated, based on the obtained results. Such initiatives, focused on artificial intelligence (AI) vision, will be a focal point for future research endeavors.

Four typical flame combustion statuses are as follows.

In Figure 3, the red arrow's direction represents the flame's orientation, while the arrow's length corresponds to the flame's height. The blue line signifies the combustion line, while the red line outlines the outer flame's edge. Figure 3a showcases a typical instance of channeling burning, characterized by localized, bright, divergent jets due to short-term material scarcity. The flame distribution indicates local channeling, particularly bright areas, and divergence. Additionally, the combustion line appears scattered in both the dry and combustion sections. Figure 3b presents a typical example of smoldering, reflecting poor MSW combustion status with substantial blackened areas in the furnace. The combustion line appears star-shaped. Figure 3c displays a typical case of partial burning, attributed to uneven material layer distribution, resulting in a dispersed yet bright flame. The combustion line assumes a curved distribution. Figure 3d depicts a typical example of normal burning, showcasing a favorable MSW combustion status. The flame remains stable, bright, and concentrated, while the combustion line maintains a straight distribution. Efforts are underway to simulate on-site expert identification methods by monitoring flame videos. With a focus on synchronous on-site monitoring of the left and right grates, there is a desire to establish a laboratory platform capable of real-time playback of flame videos. This platform aims to identify combustion statuses using an online model, facilitating the translation of laboratory research findings to industrial applications.

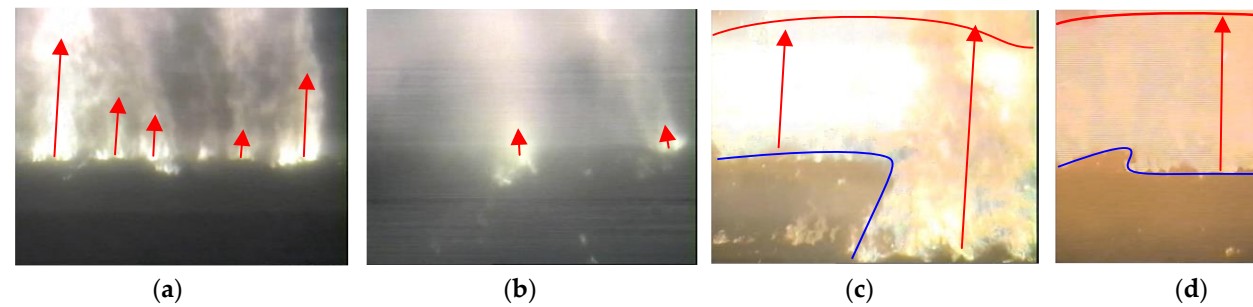

(**a**)    (**b**)    (**c**)    (**d**)

**Figure 3.** Typical flame combustion status of MSWI process: (**a**) channeling burning, (**b**) smoldering, (**c**) partial burning, (**d**) normal burning. (The red line represents the upper edge of the flame, the blue line represents the lower edge of the flame, and the red arrow represents the direction of the flame).

Absolutely, simulating the on-site experts' identification method via flame video monitoring is imperative. The aim is to realize the synchronous monitoring mode of both the left and right grates on site. Establishing a laboratory platform capable of real-time playback of flame videos and deploying an online model to identify combustion statuses

are important. This platform will serve as invaluable support in seamlessly transferring the research findings from laboratory investigations to industrial sites.

## 3. Materials and Methods

### 3.1. Materials

The plant has a capacity of 628.8 tons per day (t/d) for managing municipal solid waste. The dimensions of the grate measure 11 m in length and 12.9 m in width. The primary airflow within the system is 67,500 cubic meters per hour (m$^3$/h) at a temperature of 200 °C. The primary air is introduced into the bed through four separate sections of the grate, with each section contributing varying proportions of the total airflow: 24.31%, 43.35%, 19.27%, and 13.07%, respectively.

To capture flame videos, industrial cameras are strategically positioned at the end of the grates to enable real-time monitoring of combustion status. These cameras facilitate the transmission of MSWI flame videos via a coaxial cable. Subsequently, the videos are acquired and stored utilizing video acquisition cards. The onsite collection equipment configuration is illustrated in Figure 4.

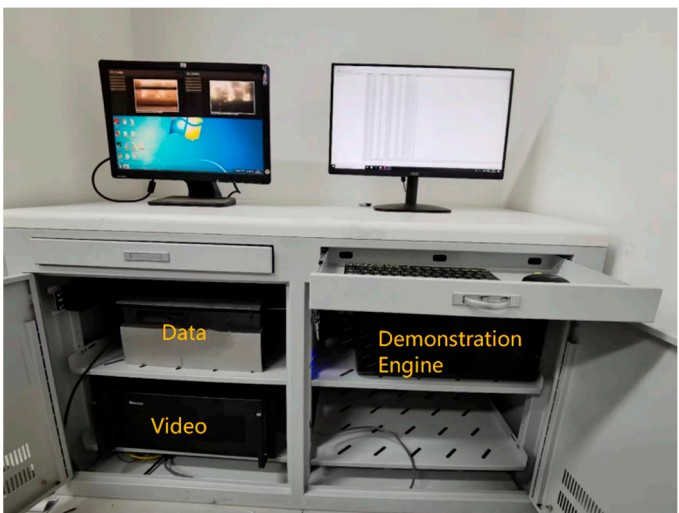

**Figure 4.** Onsite collection equipment.

We meticulously gathered flame videos from the MSWI power plant in Beijing spanning the period between November 2020 and January 2022, with video intervals ranging from 1 to 3 months. These videos comprehensively depict the combustion conditions at the MSWI plant over the entire year. Each grate's collection of flame videos has a total duration of 132 h and 30 min, recorded at a frame rate of 25 frames per second. Following a thorough screening process, we identified and isolated 54 h and 49 min of typical combustion status video clips for the left grate and 44 h and 45 min for the right grate. These carefully chosen video clips were then sampled to create a database of image frames representing typical combustion statuses. This image database served as the foundation for training the offline model. Subsequently, for the online recognition test, we utilized a distinct 2.5-h flame video collected on 21 September 2021. Notably, this particular video was not used in the offline modeling process.

### 3.2. Methods

The proposed strategy is shown in Figure 5.

Figure 5 shows that the method is mainly includes three steps: data collection and analysis, offline modeling, and online recognition. The detailed information about these steps is as follows.

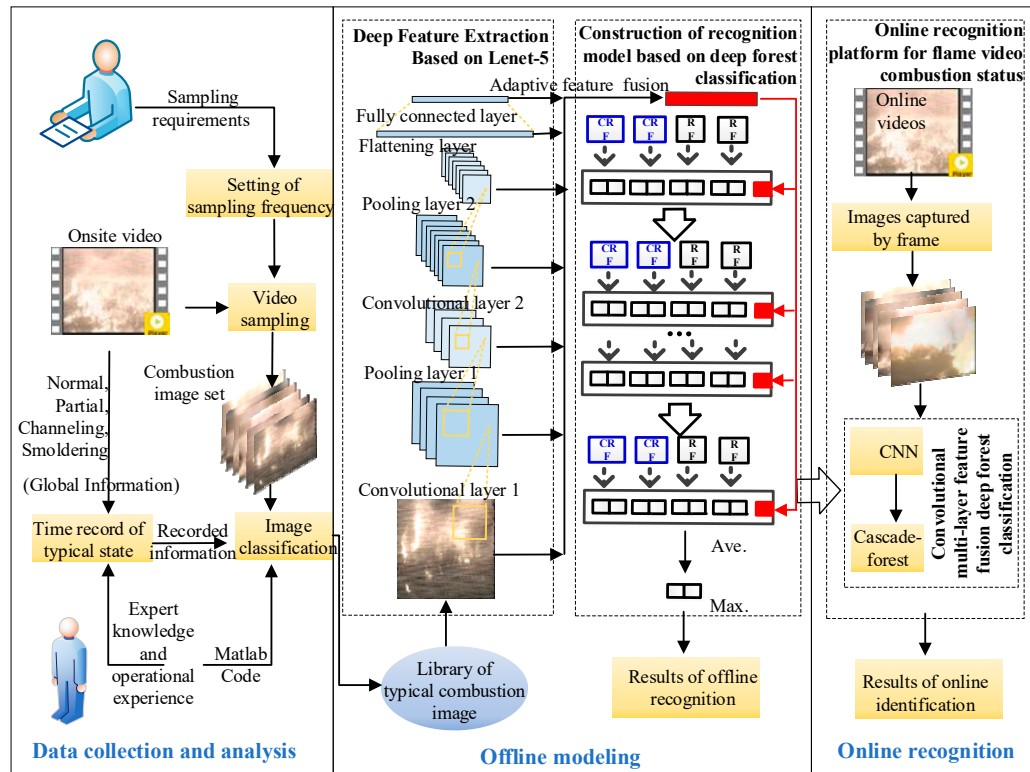

**Figure 5.** Strategy of flame video online recognition.

### 3.2.1. Data Collection and Analysis

Initially, leveraging domain expert knowledge and operational experience, the durations of typical combustion statuses within the four categories—normal, partial burning, runaway burning, and smoldering—are identified based on comprehensive insights into the global information of the flame. Subsequently, adhering to the specifications outlined by research experts, the sampling frequency is determined, enabling the extraction of a series of combustion flame images. These images serve as the foundation for constructing the offline training model. Finally, employing a DFC based on MATLAB code as the classification tool, automatic classification of typical combustion images is achieved. This classification process relies on the recorded duration information corresponding to the various typical combustion statuses, facilitating the effective categorization of these images.

### 3.2.2. Offline Modeling

The functions of each module in the offline modeling stage are as follows.

(1) Deep feature extraction module based on LeNet-5: This module is dedicated to preprocessing the training samples sourced from the library of typical combustion images. Subsequently, the LeNet-5 network undergoes training to extract profound features from flame images. The trained LeNet-5 network's output features from each layer are intelligently selected and fused adaptively, culminating in the extraction of deep fusion features inherent in flame images.

(2) Construction of recognition model based on cascaded forest: This module employs the extracted deep fusion features of flame images as the primary input for the cascaded forest. The aim is to construct a combustion status-recognition model. Through this process, the system derives the combustion status-recognition results, enabling the identification and classification of different combustion statuses.

Deep Feature Extraction Based on LeNet-5

Function description

Before inputting the flame-image dataset $\{I_n\}_{n=1}^{N}$ into LeNet-5 [12], it needs to be preprocessed to meet the network input requirements. The preprocessing operation used here is to first adjust the size of the original color flame image $I_n$ to $32 \times 32$, and then perform grayscale processing. The expression is as follows:

$$I_n^{\text{Pre}} = f_{\text{Gray}}(f_{\text{Scale}}(I_n)), \qquad n = 1, \cdots, N \tag{1}$$

where $f_{\text{Scale}}$ represents the image scaling operation and $f_{\text{Gray}}$ represents the image grayscale processing.

Then, the preprocessed image $I_n^{\text{Pre}}$ is input into the LeNet-5 network to train the ability of network to extract depth features from flame images. Figure 6 shows the model structure of LeNet-5.

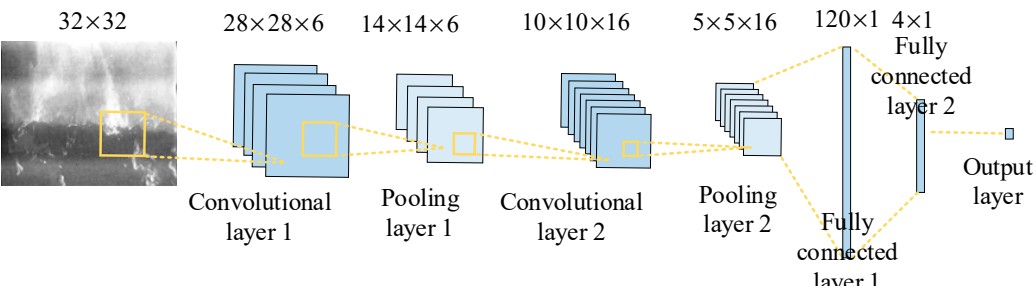

**Figure 6.** Structure of LeNet5.

As shown in Figure 6, the network mainly consists of convolutional layer 1, pooling layer 1, convolutional layer 2, pooling layer 2, fully connected layer 1, fully connected layer 2, and the output layer.

(1)  Convolutional layer 1

The convolutional layer comprises numerous convolutional kernels, and the area covered by each kernel on the input feature map is termed the receptive field. These kernels slide across the feature map with a specific step size, facilitating localized perception within their respective receptive fields. Simultaneously, every local region of the feature map shares convolutional kernel weights and bias parameters, fostering parameter sharing across the network for efficient computation.

Figure 7 is a schematic diagram of the convolution process. When the convolution kernel covers the upper left corner of the input feature map, the calculation process of the upper left corner elements $z_{11}$ of the output feature map is as follows:

$$z_{11} = a_{11} \times k_{11} + a_{12} \times k_{12} + a_{21} \times k_{21} + a_{22} \times k_{22} \tag{2}$$

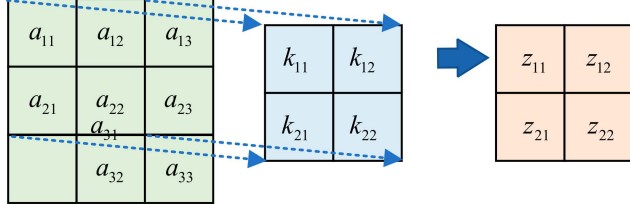

Input feature map   Convolutional kernel Output feature map

**Figure 7.** Schematic diagram of convolution process.

Afterwards, the convolution kernel slides in steps of layer 1 on the input feature map to obtain the remaining elements of the output feature map, and the convolution process is denoted as $*$.

For the convolutional layer 1 of the LeNet5 network, the input is $I_n^{\text{Pre}}$, the convolutional kernel is $K_1$ with size $5 \times 5 \times 6$, and the output net activation graph is $Z_{1,n}$ with size $28 \times 28 \times 6$. The calculation expression for $Z_{1,n}$ is as follows:

$$Z_{1,j,n} = (I_n^{\text{Pre}} * K_{1,j} + b_{1,j}), \; j = 1, 2, \ldots 6 \tag{3}$$

Then, the output feature map of convolution layer 1 $A_{1,n}$ is obtained by inputting $Z_{1,n}$ into the Tanh activation function, and its calculation expression is as follows:

$$A_{1,n} = f_{\text{tanh}}(Z_{1,n}) = \frac{e^{Z_{1,n}} - e^{-Z_{1,n}}}{e^{Z_{1,n}} + e^{-Z_{1,n}}} \tag{4}$$

where $f_{\text{tanh}}(\cdot)$ represents the Tanh activation function.

(2)  Pooling layer 1

The pooling layer, also known as the downsampling layer, is used to reduce overfitting in the network by sparsely processing the feature maps. The pooling kernel of the pooling layer only consists of a framework and does not have specific parameters. Similar to the convolutional layer, the pooling kernel slides over the input feature maps with a certain stride and performs either max pooling or average pooling on the feature maps. Max pooling takes the maximum feature value within the pooling region, while average pooling calculates the average value of the feature maps within the pooling region. Compared to max pooling, average pooling helps to preserve the overall trend of the flame image and retain more background information, which is important for flame images. In this case, average pooling is used, and its calculation expression is as follows:

$$A_{2,n} = mean(A_{1,n}, K_2) \tag{5}$$

where $mean(\cdot)$ represents the mean function of the matrix, $K_2$ ($2 \times 2 \times 6$) is the pooling kernel used to determine the size of the mean matrix, and $A_{2,n}$ (size $14 \times 14 \times 6$) is the pooling layer output feature map.

(3)  Convolutional layer 2

When the input feature map is multi-channel, the schematic diagram of the convolution process is shown in Figure 8.

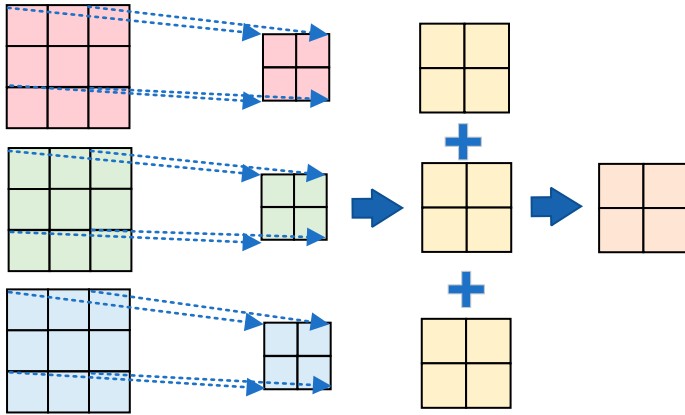

**Figure 8.** Process of multichannel convolution.

As shown in Figure 7, the number of channels in the convolutional kernel is the same as the number of channels in the input feature map. The number of output feature map channels is the same as the number of convolutional kernels. The multi-channel convolution result is the sum of the convolution operations performed on each channel

of the input feature map and each channel of the convolution kernel. The multi-channel convolution operation is denoted as $\otimes$.

For the convolutional layer 2 of the LeNet5 network, the input is $A_{2,n}$, the convolutional kernel is $K_3$ with size $5 \times 5 \times 6 \times 16$, and the output net activation graph is $Z_{2,n}$ with size $10 \times 10 \times 16$. The calculation expression for $Z_{2,n}$ is as follows:

$$Z_{2,m,n} = (A_{2,n} \otimes K_{3,m}) + b_{2,m}, \; m = 1, 2, \ldots 16 \tag{6}$$

Then, $Z_{2,n}$ is input into the Tanh activation function $f_{\text{tanh}}(\cdot)$ to obtain the output feature map $A_{3,n}$ of convolution layer 2, which is calculated as follows:

$$A_{3,n} = f_{\text{tanh}}(Z_{2,n}) \tag{7}$$

(4)   Pooling layer 2

Consistent with pooling layer 1, average pooling is used here. Its calculation expression is as follows:

$$A_{4,n} = mean(A_{3,n}, K_4) \tag{8}$$

where $K_4$ ($2 \times 2 \times 16$) is the pooling core and $A_{4,n}$ ($5 \times 5 \times 16$) is the pooling layer output feature map.

(5)   Fully connected layer 1

The function of the fully connected layer is to map the learned features to the sample space. For the fully connected layer 1 of LeNet5, the processing process for $A_{4,n}$ is as follows:

$$z_{3,n} = (A_{4,n} \otimes K_5) + b_{3,n}, \; n = 1, 2, \ldots, 120 \tag{9}$$

where the size of $K_5$ is $5 \times 5 \times 16 \times 120$ and the size of $z_{3,n}$ is $1 \times 120$.

Then, $z_{3,n}$ is input into the Tanh activation function to obtain the output feature map $a_{5,n}$ of fully connected layer 1, which is calculated as follows:

$$a_{5,n} = f_{\text{tanh}}(z_{3,n}) \tag{10}$$

(6)   Fully connected layer 2

For the fully connected layer 2 of LeNet5, the processing process for $a_{5,n}$ is as follows:

$$z_{4,n} = k_6 a_{5,n} + b_4 \tag{11}$$

where the size of $k_6$ and $z_{4,n}$ are $120 \times 4$ and $1 \times 4$.

Then, $z_{4,n}$ is input into the Tanh activation function to obtain the output $a_{6,n}$ of fully connected layer 1, which is calculated as follows:

$$a_{6,n} = f_{\text{tanh}}(z_{4,n}) \tag{12}$$

(7)   Output layer

Finally, the output of fully connected layer 2 $a_{6,n}$ is processed by Softmax to obtain the probability values $\hat{y}_n$ of the input image belonging to labels. The expression is as follows:

$$\hat{y}_{t,n} = \frac{e^{a_{6,t,n}}}{\sum\limits_{i=1}^{T} e^{a_{6,i,n}}}, \; t = 1, 2, \cdots, T \tag{13}$$

where $T = 4$ represents the number of classes and $e$ represents the base of the natural logarithm.

Parameter learning process

The parameters that need to be learned mainly include the weight matrices $K_1$, $K_3$, $K_5$, and $k_6$ of convolution layer 1, convolution layer 2, fully connected layer 1, and fully connected layer 2, as well as bias parameters $b_1$, $b_2$, $b_3$, and $b_4$.

LeNet5 uses the gradient descent algorithm to calculate the backpropagation of errors, and then uses the SGD algorithm to update the network parameters. When selecting the loss function, the mean squared error (MSE) is widely used, due to its intuitive, easy-to-compute, and smooth characteristics. So, the used loss function is the MSE loss, which is expressed as:

$$C = \frac{1}{2} \|a_{6,n} - y_n\|_2^2 \tag{14}$$

where $\| \cdot \|_2$ represents the L2 norm.

The specific process of deriving network node gradients from backward to forward in the backpropagation algorithm is as follows.

(1) Parameter updated for fully connected layer 2.

First, the error $\delta_6$ of the loss function on the $z_{4,n}$ output layer of fully connected layer 2 is calculated as follows:

$$\delta_6 = \frac{\partial C}{\partial z_{4,n}} = \frac{\partial C}{\partial a_{6,n}} \cdot \frac{\partial a_{6,n}}{\partial z_{4,n}} = (a_{6,n} - y_n) \odot f'(z_{4,n}) \tag{15}$$

where $\odot$ represents the Hadmard product and the expression for $f'(z_{4,n})$ is

$$f'(z_{4,n}) = 1 - (a_{6,n})^2 \tag{16}$$

Then, $\delta_6$ is used to calculate the gradient of the loss function on the parameters of the layer:

$$\frac{\partial C}{\partial k_6} = \frac{\partial C}{\partial a_{6,n}} \cdot \frac{\partial a_{6,n}}{\partial k_6} = \delta_6 (a_{5,n})^T \tag{17}$$

$$\frac{\partial C}{\partial b_4} = \delta_6 \tag{18}$$

Finally, the error $\delta_6$ is used to calculate the gradient of the loss function on the parameters of the layer:

$$\frac{\partial C}{\partial k_6} = \frac{\partial C}{\partial a_{6,n}} \cdot \frac{\partial a_{6,n}}{\partial k_6} = \delta_6 (a_{5,n})^T \tag{19}$$

$$\frac{\partial C}{\partial b_4} = \delta_6 \tag{20}$$

(2) Parameter updated for fully connected layer 1.

First, the error recurrence formula between adjacent layers is used to find $\delta_5$:

$$\delta_5 = (k_6)^T \delta_6 \odot f'(z_{3,n}) = (k_6)^T \delta_6 \odot [1 - (a_{5,n})^2] \tag{21}$$

Then, the error is used to calculate the gradient of the loss function for the layer parameters:

$$\frac{\partial C}{\partial K_5} = \frac{\partial C}{\partial a_{5,n}} \cdot \frac{\partial a_{5,n}}{\partial K_5} = \delta_5 (A_{4,n})^T \tag{22}$$

$$\frac{\partial C}{\partial b_3} = \delta_5 \tag{23}$$

(3) There is no parameter update for pooling layer 2, but intermediate layer error $\delta_4$ needs to be passed:

$$\delta_4 = (K_5)^T \delta_5 \tag{24}$$

(4) Parameter updated for convolutional layer 2.

First, the error recurrence formula is used between adjacent layers to find $\delta_3$:

$$\delta_3 = \text{upsample}(\delta_4) \odot f'(z_{3,n}) = \text{upsample}(\delta_4) \odot [1 - (a_{5,n})^2] \tag{25}$$

where upsample$(\cdot)$ represents the upsampling operation.

The specific processing process is as follows.

First, $\delta_3$ is restored to the size before pooling; then, due to average pooling, the elements of $\delta_3$ are averaged and restored to the submatrix. Error $\delta_3$ is used to calculate the gradient of the loss function on the parameters of this layer:

$$\frac{\partial C}{\partial K_3} = \delta_3 * A_{2,n} \tag{26}$$

$$\frac{\partial C}{\partial b_2} = \sum_{u=1}^{U} \sum_{v=1}^{V} \delta_3^{u,v} \tag{27}$$

(5) There is no parameter update for pooling layer 1, but intermediate layer error $\delta_2$ needs to be passed:

$$\delta_2 = \delta_3 * \text{ROT180}(K_3) \tag{28}$$

(6) Convolutional layer 1 parameter update.

The error recurrence formula between adjacent layers is used to calculate $\delta_1$:

$$\delta_1 = \text{upsample}(\delta_2) \odot f'(z_{1,n}) = \text{upsample}(\delta_2) \odot [1 - (A_{1,n})^2] \tag{29}$$

where $\delta_1$ is used to calculate the gradient of the loss function on the parameters of this layer:

$$\frac{\partial C}{\partial K_1} = \delta_1 * I_n^{\text{Pre}} \tag{30}$$

$$\frac{\partial C}{\partial b_1} = \sum_{u=1}^{U} \sum_{v=1}^{V} \delta_1^{u,v} \tag{31}$$

The SGD algorithm is used to update the parameter values, as shown:

$$\theta_p = \theta_{p-1} - \alpha \nabla_p, \ p = 1, \cdots, P \tag{32}$$

where $\theta_p$ represents the network parameters at the $p$-th iteration, $\alpha$ is the learning rate, $P$ represents the total number of iterations of network training, and $\nabla_p$ represents the parameter gradient calculated during the $p$-th backpropagation.

After completing the training of the LeNet-5 network, it has the ability to extract depth features from flame images. In order to increase the diversity and complementarity of features and effectively characterize flame images, adaptive selection fusion processing between multiple layers of features is performed on the output feature maps of each layer of the LeNet-5 network. The specific steps are as follows:

Step (1): The output feature $[s_n^1, s_n^2, s_n^3, s_n^4, s_n^5, s_n^6]$ of each layer is extracted and saved;

Step (2): $[s_n^1, s_n^2, s_n^3, s_n^4, s_n^5, s_n^6]$ is flattened to obtain the one-dimensional vector form $[s_n^1, s_n^2, s_n^3, s_n^4, s_n^5, s_n^6]$ of each layer;

Step (3): The features of each layer in different combinations are combined and concatenated;

Step (4): Each of the combined features is input into the recognition model to construct different recognition models and the performances of each recognition model are compared;

Step (5): The feature combination corresponding to the best performance recognition model is used as the final flame image deep fusion feature $s_n^{\text{Fusion}}$.

Construction of Recognition Model Based on Deep Forest Classification (DFC)

To enhance the model's performance, the DFC's multi-granularity scanning module [24] has been excluded, utilizing solely the CF module for constructing the combustion status-recognition model. Within each CF layer, the base learners employed are RF and CRF. The structural configuration of the recognition model based on the CF is depicted in Figure 9.

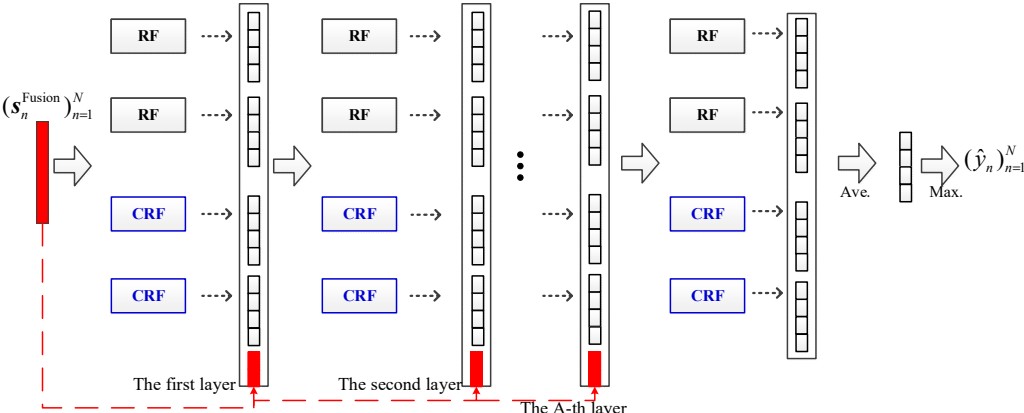

**Figure 9.** Structure of CF based recognition model.

In the DFC model, each layer of CF contains 2 RFs and 2 CRFs for cascade learning. The CF layer model is constructed in terms of stack ensemble. $(s_n^{\text{Fusion}})_{n=1}^N$ is input into CF to construct a recognition model. Except for the first CF layer, where $(s_n^{\text{Fusion}})_{n=1}^N$ is directly used as the input feature of each forest learner, subsequent CF layers need to concatenate the class distribution vector output from the previous layer with $(s_n^{\text{Fusion}})_{n=1}^N$ as the input of this CF layer to effectively prevent overfitting of the stack strategy. The number of CF layers is adaptively adjusted through cross validation.

RF Algorithm

RF is an ensemble model based on bagging method, which is constructed with decision trees (DTs). It was proposed by Breiman et al. [41].

Bootstrap is used to randomly sample training set $\dot{S} = \{(s_i, y_i), i = 1, 2, \cdots I\}$. The generation process of RF training subset $G$ can be described as follows:

$$\{(g^{c,M^c}, y^c)_1^i\}_{i=1}^I = f_{\text{Gini}}(f_{\text{Bootstrap}}(\dot{S}, G), R^c) \tag{33}$$

where $\{(g^{c,M^c}, y^c)_1^i\}_{i=1}^I$ represents the $c$-th training subset, $f_{\text{Gini}}(\cdot)$ represents a random subspace function, $f_{\text{Bootstrap}}(\cdot)$ represents the bootstrap function, and $r = 1, \cdots, R^c$, $R^c$ represents the number of features selected for the $c$-th training subset in the forest, $R^c << R$.

By using the above function $C$ times, the training set of RF can be obtained:

$$\left.\begin{matrix} \dot{S} \\ C \end{matrix}\right\} \Rightarrow \begin{cases} \{(g^{1,R^1}, y^1)_1^i\}_{i=1}^I \\ \cdots \\ \{(g^{c,R^c}, y^c)_1^i\}_{i=1}^I \\ \cdots \\ \{(g^{1,R^C}, y^C)_1^i\}_{i=1}^I \end{cases} \tag{34}$$

where $C$ represents the number of bootstraps and the number of DT in RF.

DTs are constructed in the RF model using the training subsets. The process is described using $\{(g^{c,M^c}, y^c)^i_1\}^I_{i=1}$ as an example. Based on the Gini index criterion, the best segmentation feature number $R^c_{\text{sel}}$ and segmentation point $s$ is found:

$$(R^c_{\text{sel}}, s) = \text{argmin}[\frac{y^c_{P_{\text{Left}}}}{y^c}\text{Gini}(y^c_{P_{\text{Left}}}) + \frac{y^c_{P_{\text{Right}}}}{y^c}\text{Gini}(y^c_{P_{\text{Right}}})] \tag{35}$$

$$\text{Gini}(\cdot) = \sum_{c_p=1}^{C_p} p_{c_P}(1 - p_{c_p}) = 1 - \sum_{c_p=1}^{C_p} p_{c_P}{}^2 \tag{36}$$

$$s.t. \begin{cases} P_{\text{Left}} > \theta_{\text{Forest}} \\ P_{\text{Right}} > \theta_{\text{Forest}} \\ \text{Gini}(y^c_{P_{\text{Left}}}) > 0 \\ \text{Gini}(y^c_{P_{\text{Right}}}) > 0 \end{cases}$$

where $c_P$ represents class $c_P$ in dataset label $y$, $c_P \in 1, \cdots, C_P$; $p_{c_P}$ represents the proportion of $c_P$ to the total number of labels; $\text{Gini}(\cdot)$ represents the Gini index; $\theta_{\text{Forest}}$ represents the threshold for the number of samples contained in the leaf node; $y^c_{P_{\text{Left}}}$ and $y^c_{P_{\text{Right}}}$ represent the label values corresponding to the samples divided into left and right nodes in the $c$-th training subset, respectively.

Based on the above criteria, the optimal variable number and segmentation point value are found by first traversing all input features. The input feature space is divided into left and right regions. Then, the above process is repeated for each region until the number of samples contained in the leaf node is less than $\theta_{\text{Forest}}$, or the Gini index of the samples in the leaf node is 0. Finally, the input feature space is divided into $Q$ regions. To construct a classification tree model, the following functions is defined:

$$\Gamma^c(\cdot) = \sum_{q=1}^{Q} p^q_c \Lambda(p^{c,R^c} \in G_q) \tag{37}$$

where

$$p^q_c = [p_1, \cdots, p_{c_p}, \cdots, p_{C_p}]^T(y^c_{N_{R_q}} \in G_q, N_{G_q} \leq \theta_{\text{Forest}}) \tag{38}$$

where $N_{G_q}$ represents the number of training samples contained in region $G_q$; $y^c_{N_{R_q}}$ represents the label vector corresponding to the sample features in region $G_q$; $p^q_c$ represents the predicted result of the final output of $G_q$; and to indicate the function $\Lambda(\cdot)$, when $p^{c,R^c} \in G_q$, $\Lambda(\cdot) = 1$, otherwise $\Lambda(\cdot) = 0$.

The RF model obtained by repeating the above step $C$ times:

$$F_{\text{RF}}(\cdot) = \text{arg}(\max_{c_P} \frac{1}{C}\sum_{c=1}^{C}\Gamma^c(\cdot)) \tag{39}$$

CRF Algorithm

The difference between CRF and RF is that the former randomly selects the value of a certain feature as a splitting node in the complete feature space, while the latter selects the splitting node in the bootstrap random feature subspace through Gini coefficients. Correspondingly, the CRF model is represented as $F_{\text{CRF}}(\cdot)$.

Output of DFC

Each layer of CF uses 2 $F_{\text{RF}}(\cdot)$ and 2 $F_{\text{CRF}}(\cdot)$ for cascade learning. The stack ensemble method is used to construct the CF model. For input $s^{\text{Fusion}}_n$, the last layer of CF will output

the $4C_p$-dimensional class distribution vector $Res_n = [\boldsymbol{r}_1^{\mathrm{RF}}, \boldsymbol{r}_2^{\mathrm{RF}}, \boldsymbol{r}_1^{\mathrm{CRF}}, \boldsymbol{r}_2^{\mathrm{CRF}}]$. The average and maximum criteria are used to obtain the recognition result $\hat{y}_n$,

$$\hat{y}_n = \max[\frac{1}{4} \times Res_n] \tag{40}$$

For feature $(\boldsymbol{s}_n^{\mathrm{Fusion}})_{n=1}^N$, the final combustion status-recognition result $(\hat{y}_n)_{n=1}^N$ can be obtained.

In the recognition module, the number of decision trees $C$ (we denote it as Tree_Number later) and the minimum number of leaf nodes $\theta_{\mathrm{Forest}}$ (we denote it as Mini_Samples later) in each forest need to be determined, while other parameters remain default.

### 3.2.3. Online Recognition

In the online recognition stage for combustion status, the process begins with capturing flame videos, which are then subjected to image preprocessing. Following this, the preprocessed images undergo deep feature extraction through the LeNet-5 network. Subsequently, the output features from the intermediate layers of LeNet-5 are intelligently fused, based on an adaptive selection fusion mechanism. These fused features serve as the input for the DFC model, facilitating the recognition of combustion statuses. Ultimately, this sequence culminates in obtaining the online recognition result.

The schematic diagram of on-site layout of the online identification is shown in Figure 10.

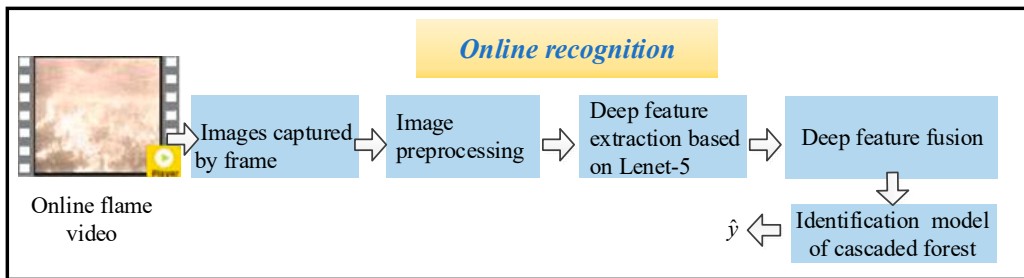

**Figure 10.** Process of online identification.

## 4. Results and Discussion

### 4.1. Data Collection and Analysis Results

The flame-image dataset utilized in this experiment originates from an MSWI plant located in Beijing. To ensure comprehensive coverage despite the limited field of view of industrial cameras, each end of the left and right grates onsite is equipped with a dedicated camera for flame video collection. The process of handling the collected videos involves initially selecting typical combustion status segments within the flame videos. Upon collection of flame videos from both the left and right grates onsite, the initial step involves the removal of fragments depicting unclear combustion statuses. Following this, the remaining video segments are classified according to the combustion status classification standard illustrated in Figure 2. These classified video segments are subsequently sampled at a consistent rate of 1 frame per minute utilizing a MATLAB program, resulting in the extraction of flame-image frames. Consequently, the total count of typical combustion status images obtained from the left and right furnace bars is 3289 and 2685, respectively. For a detailed breakdown of each typical combustion status, please refer to Table 1.

**Table 1.** Flame-image dataset.

| Grate | Amount | Normal | Partial | Channeling | Smoldering | Size |
|-------|--------|--------|---------|------------|------------|------|
| Left | 3289 | 655 | 1176 | 1044 | 414 | $720 \times 576$ |
| Right | 2685 | 564 | 1002 | 534 | 585 | $720 \times 576$ |

### 4.2. Offline Modeling Results

4.2.1. Evaluation Indices

Table 2 shows the confusion matrix of the classification results.

**Table 2.** Confusion matrix of classification result.

| True Situation | Prediction Result | |
| --- | --- | --- |
| | **Positive** | **Negative** |
| Positive | TP | FN |
| Negative | FP | TN |

In Table 2, the directional columns within the confusion matrix denote the prediction outcomes, whereas the directional rows signify the actual results. By analyzing the confusion matrix, it becomes evident where the model tends to misclassify during predictions. To assess the model's performance, evaluation indices such as accuracy, precision, and recall are employed. They are calculated as follows,

$$\text{Accuracy} = \frac{\text{TP} + \text{TN}}{\text{TP} + \text{TN} + \text{FP} + \text{FN}} \tag{41}$$

$$\text{Precision} = \frac{\text{TP}}{\text{TP} + \text{FP}} \tag{42}$$

$$\text{Recall} = \frac{\text{TP}}{\text{TP} + \text{FN}} \tag{43}$$

4.2.2. Result of Method Comparison

The training, validation and testing datasets are divided according to the ratio of 2:1:1 of the samples. In order to verify the superiority of the proposed method, it is compared with the classical CNN method. The settings of parameter are shown in the Table 3.

**Table 3.** Settings of CNNs parameter.

| Methods | Settings of Parameter | | |
| --- | --- | --- | --- |
| | **Epochs** | **Learning_Rate** | **Batch_Size** |
| VGGnet | 74 | 0.01 | 64 |
| Mobilenet | 90 | 0.045 | 64 |
| Densenet | 90 | 0.1 | 16 |
| EfficientNet | 90 | 0.256 | 64 |
| LeNet-5 (Left) | 28 | 0.01 | 100 |
| LeNet-5 (Right) | 39 | 0.01 | 100 |
| Regnet | 90 | 0.1 | 64 |

The parameters of DFC are set as follows: Tree_Number = 30, Mini_Samples = 5. At the same time, the cascade layer is set to adaptively adjust using cross-validation results.

Tables 4 and 5, respectively, show the experimental results of the recognition models constructed by each method based on left- and right-grate flame images.

From the comparative experimental results presented above, it is evident that, despite being the most fundamental network, LeNet-5 outperforms other CNN models in flame combustion status recognition with fewer training epochs. Interestingly, even without a multi-granularity scanning module, the recognition model constructed with DFC manages to achieve commendable recognition results. Building upon this insight, this study extracts depth features from flame images using LeNet-5 and dynamically selects and merges the

intermediate layer features as input for constructing a recognition model with DFC. The experimental findings demonstrate a substantial enhancement in recognition performance when compared to the original recognition models employing LeNet-5 and DFC. This shows LeNet-5's proficiency in effectively extracting deep flame-image features. Additionally, the adaptive selection and fusion of features from each intermediate layer exhibit stronger complementarity. Consequently, upon integrating with DFC, the model's recognition efficacy using adaptive selection features has a remarkable improvement.

**Table 4.** Comparative experimental results of left grate.

| Methods | Evaluation Index | | |
|---|---|---|---|
| | Accuracy | Precision | Recall |
| VGGnet | 0.36893 | 0.09223 | 0.25 |
| Mobilenet | 0.81553 | 0.80217 | 0.75971 |
| Densenet | 0.83252 | 0.85054 | 0.78825 |
| EfficientNet | 0.55097 | 0.6452 | 0.60138 |
| Regnet | 0.7185 | 0.7124 | 0.7248 |
| LeNet-5 | 0.8990 | 0.8986 | 0.8929 |
| DFC | 0.8832 | 0.8576 | 0.9022 |
| Ours | 0.9380 | 0.9182 | 0.9507 |

**Table 5.** Comparative experimental results of right grate.

| Methods | Evaluation Index | | |
|---|---|---|---|
| | Accuracy | Precision | Recall |
| VGGnet | 0.36418 | 0.09104 | 0.25 |
| Mobilenet | 0.77313 | 0.80396 | 0.75911 |
| Densenet | 0.87164 | 0.86668 | 0.88562 |
| EfficientNet | 0.77313 | 0.77245 | 0.77835 |
| Regnet | 0.8269 | 0.8211 | 0.8295 |
| LeNet-5 | 0.9151 | 0.9122 | 0.9149 |
| DFC | 0.8942 | 0.8848 | 0.9001 |
| Ours | 0.9508 | 0.9456 | 0.9541 |

4.2.3. Results of Offline Recognition

As shown in Figure 11, the training process of the left- and right-grate flame images is based on LeNet-5.

As illustrated in Figure 11, the loss curve exhibits an initial decrease followed by a gradual stabilization, indicating convergence. Similarly, the accuracy curve displays an initial ascent followed by a steady level, affirming that the models have converged and possess a robust capability to extract deep features from flame images.

Following the training of the LeNet-5 network, the extracted intermediate layer features undergo an adaptive selection and fusion process. Subsequently, these fused features are utilized as inputs for constructing a recognition model within the DFC framework. The comparison among various recognition models yields the ultimate multi-layer feature adaptive selection fusion outcomes. The fusion recognition results for each layer of the left- and right-grate flame images are detailed in Tables 6 and 7, respectively.

Table 6 shows that for the flame image of the left grate, the best recognition result can be achieved by fusing the depth features of the flame image extracted from layers 4–6. Table 7 shows that for the flame image of the right grate, the best recognition result can

be achieved by fusing the depth features of the flame image extracted from layers 3–6. The results in multi-layer feature adaptive selection of the left grate and the right grate indicate that there are certain differences in the quality of left- and right-grate flame images. Therefore, it is necessary to construct recognition models based on left- and right-grate flame images separately.

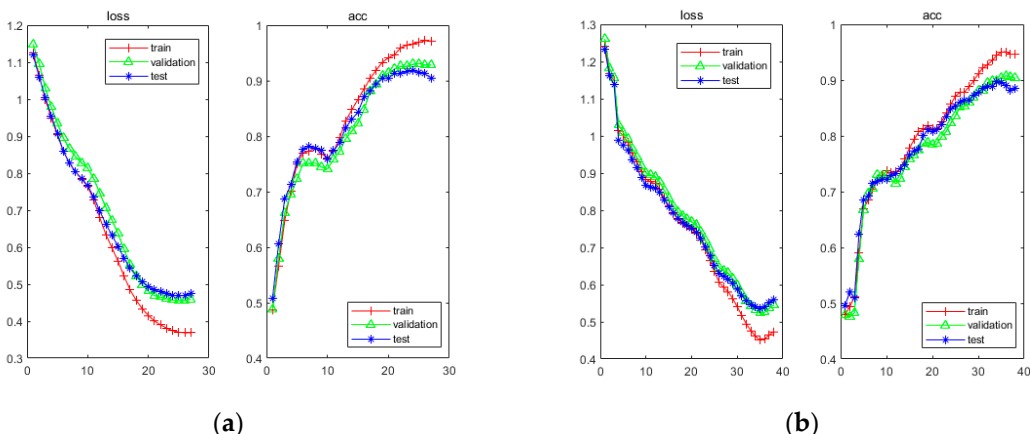

(a)          (b)

**Figure 11.** Training process of LeNet-5: (**a**) left grate; (**b**) right grate.

**Table 6.** Fusion results of multilayer feature adaptive selection for left grate.

| Layers | Evaluation Index | | |
| --- | --- | --- | --- |
| | Accuracy | Precision | Recall |
| 1–6 | 0.8917 | 0.8694 | 0.9174 |
| 2–6 | 0.9124 | 0.8894 | 0.9265 |
| 3–6 | 0.9039 | 0.8800 | 0.9238 |
| 4–6 | 0.9380 | 0.9182 | 0.9507 |
| 5–6 | 0.9112 | 0.8942 | 0.9143 |
| 5 | 0.8966 | 0.8743 | 0.9006 |

**Table 7.** Fusion results of multilayer feature adaptive selection for right grate.

| Layers | Evaluation Index | | |
| --- | --- | --- | --- |
| | Accuracy | Precision | Recall |
| 1–6 | 0.9121 | 0.9028 | 0.9263 |
| 2–6 | 0.9359 | 0.9279 | 0.9470 |
| 3–6 | 0.9508 | 0.9456 | 0.9541 |
| 4–6 | 0.9329 | 0.9305 | 0.9322 |
| 5–6 | 0.9091 | 0.9111 | 0.9050 |
| 5 | 0.8972 | 0.8967 | 0.8950 |

4.2.4. Sensitivity Analysis of Hyperparametric

Taking the model built by the left grate as an example, the sensitivity analysis of Tree_Number and Mini_Samples are shown in Figures 12 and 13.

As shown in Figure 12, the model performance gradually improves with the increase in Tree_Number. When the Tree_Number increases from 1 to 10, the model performance improves significantly. Afterwards, with the continuous increase in Tree_Number, the model performance slight fluctuates within a certain range.

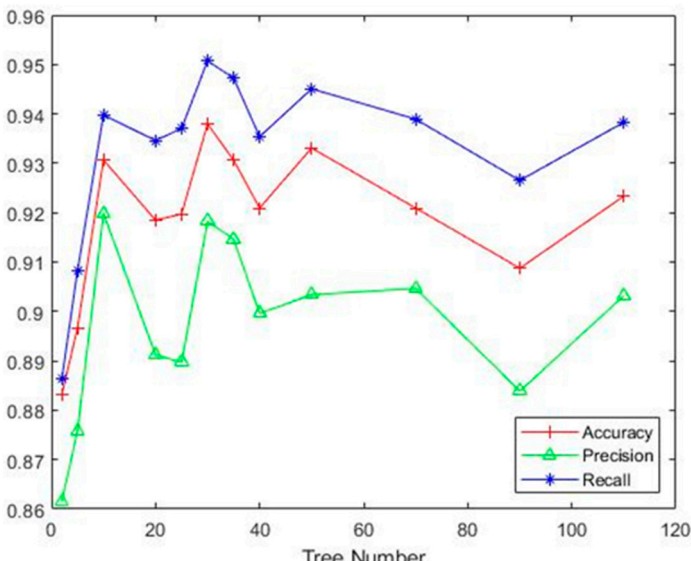

**Figure 12.** Sensitivity analysis curve of Tree_Number.

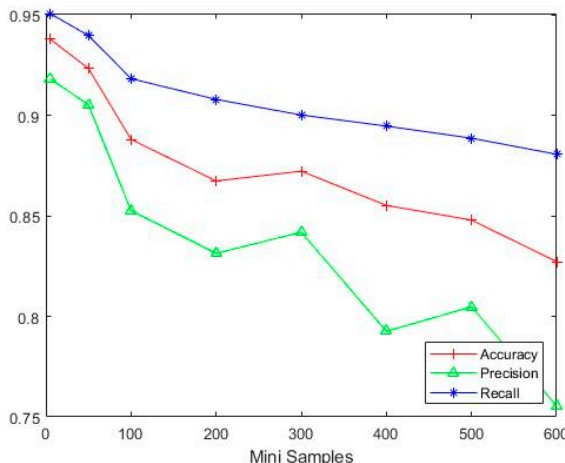

**Figure 13.** Sensitivity analysis curve of Mini_Samples.

As shown in Figure 13, with the increase in Mini_Samples, the performance of the recognition model gradually decreases.

### 4.3. Online Recognition Results

This article designs an MSWI process data monitoring system based on MATLAB APP designer. In the system, process data are directly displayed on the interface through corresponding tags, and flame video is used to recognize the combustion status by using the offline modeled recognition model. The recognition results are displayed above the flame video. The sampling frequency of process data is once per second. The sampling frequency of video can be set, with the unit being minutes. Figure 14 shows the online identification results of flames in different combustion status for the designed system.

Figure 14 illustrates the system devised in this article, capable of visually presenting process data and flame videos. It successfully accomplishes the recognition of online flame videos utilizing the designed recognition algorithm. This system effectively eliminates the instability in recognition stemming from manual experience, laying the foundation for advanced research in intelligent control.

From Figure 14, it is evident that the software not only presents the current combustion status-recognition results of the flame video but also assigns a probability value. This additional detail is reasonable, due to the complexity of onsite combustion status, where distinct

boundaries between the four categories of combustion status might not always be clear. In situations involving transitional or coupled phases of different combustion statuses, the probability representation mode is employed. This approach enables operators in practical MSWI plants to judge the confidence level of recognition results, providing a valuable reference for adjusting control strategies. It offers operators a clearer understanding of the degree of coupling within the current combustion status. Furthermore, this enhances the need for a dynamic recognition method based on contextual image correlation in future advancements.

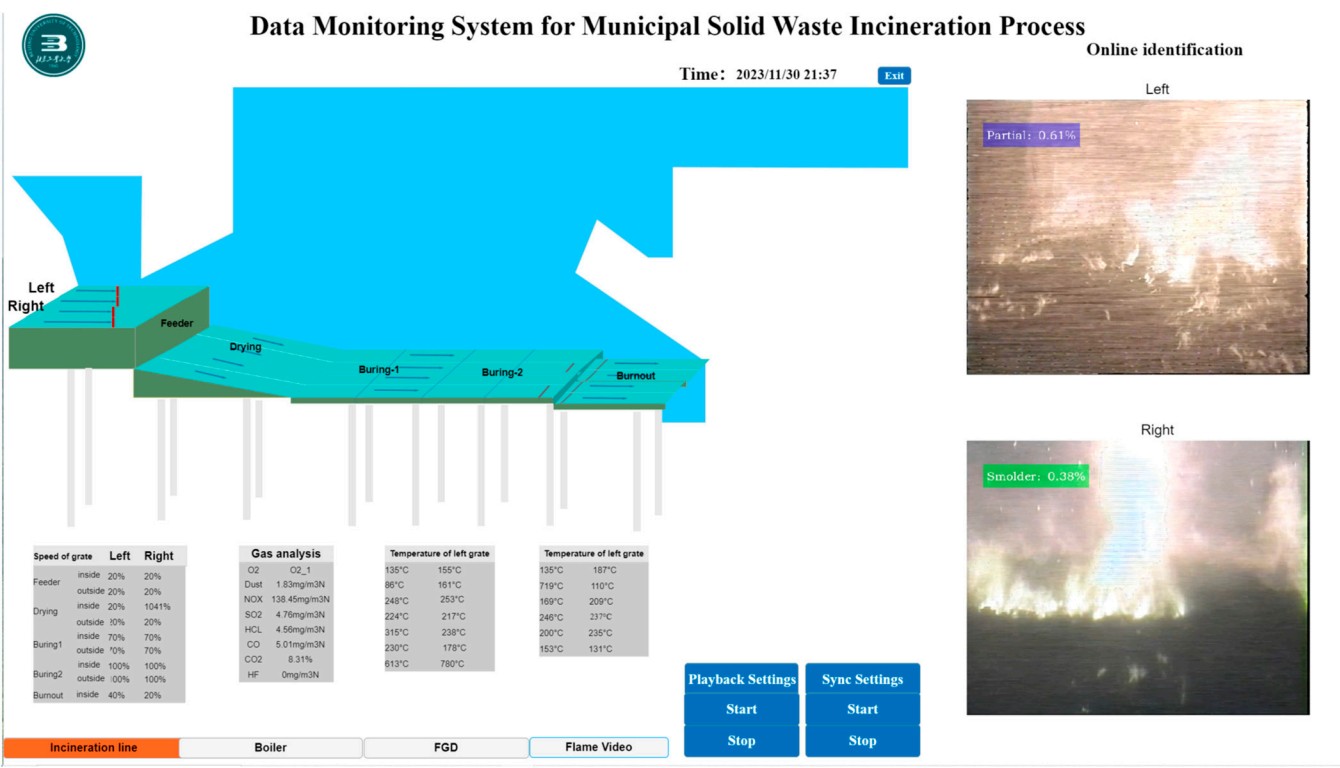

**Figure 14.** Online identification results of designed system.

The hardware configuration used for building the model included an Intel® Core™ i9-11900K CPU, 32 GB of RAM (Santa Clara, CA, USA), and an NVIDIA GeForce RTX3060Ti GPU (Santa Clara, CA, USA). The integrated development environment was MATLAB 2021b. The time required for the trained offline recognition model to analyze and recognize a flame image averaged approximately 0.174 s. During the online recognition process, flame images were sampled every minute to assess their combustion status. Given the relatively slow change in combustion status, this recognition speed effectively met the requirements for real-time online recognition.

### 4.4. Comprehensive Analysis

For the method proposed in this article, there are some limitations in each stage, as follows.

(1) In the data collection and analysis stage, the selection of typical video clips was meticulously performed through expert labeling, excluding videos with severe combustion status coupling. Consequently, some unclear videos were not utilized in this study. In future research, these video clips might undergo denoising techniques before expert labeling is applied. Furthermore, the sampled video frames contain visual representations of various process data. For instance, Wang [42] utilized CCD radiation energy images to reconstruct the temperature distribution within the incineration system. Similarly, He et al. [43] measured flame radiation spectra to acquire temperature and emission rates of the burning flames, and Xie et al. [44] predicted calorific value by

employing Yolov5 to identify waste types in images. Subsequent efforts will focus on integrating these images with the corresponding process data.

(2) In the offline modeling phase, the CNN-based feature extraction predominantly emphasizes local flame image features, potentially neglecting key global features essential for comprehensively observing complex combustion status within the MSWI process. In our prior investigations, we explored a combustion status-recognition technique employing Vision Transformer-IDFC [45], leveraging the transformer's self-attention mechanism to extract significant global features from flame images, resulting in commendable recognition outcomes. Consequently, addressing the identification of complementary features and the elimination of redundant ones becomes necessary, accomplished by employing feature selection methods that aim for maximal correlation and minimal redundancy. Additionally, optimizing classifier hyperparameters concurrently with those used in feature engineering can improve the generalization performance of the recognition model. To tackle this challenge, we aim to employ intelligent optimization algorithms inspired by biological intelligence, like genetic algorithms, differential evolution, and particle swarm optimization. Nevertheless, these approaches may introduce computational complexities with long running times. As a remedy, optimization using proxy models will be employed to address this new challenge.

(3) In the online recognition phase, we capture flame video frames at regular intervals and employ offline constructed recognition model for identification. The obtained recognition results are then fed back into the online recognition system, displaying them on the desktop. However, this process inherently employs a single-image identification method, lacking consideration for the temporal relationships and causal changes between image sequences over time. Flame videos, as a form of streaming data, encapsulate both spatial information within frames and temporal information between frames. Regrettably, the current recognition system neglects this temporal dimension. Techniques from other domains specializing in stream image mining and analysis, such as active learning with expert input [46], real-time video stream analytics [47], and streaming deep neural networks (DNN) [48], can be integrated. This enhancement would facilitate applications in the actual MSWI process, paving the way for intelligent control based on AI vision.

## 5. Conclusions

In response to the practical need for reducing emissions and energy consumption in the treatment of MSW using a grate furnace within the MSWI process, we developed an online combustion status-recognition method. Based on a database of flame images depicting typical combustion statuses, our approach involves utilizing convolutional multi-layer feature fusion and DFC. Initially, a LeNet-5 network undergoes training to extract deep features from flame images across various typical combustion statuses. These extracted deep features are selectively fused using a multi-layer feature adaptive selection method, forming a comprehensive representation of flame combustion status. Subsequently, the fused depth features are fed into the DFC to establish an offline recognition model. Ultimately, this model facilitates the realization of online flame video recognition.

This study presents several notable advantages: (1) Advanced combination: It marks the first time of successfully combining LeNet-5 and DFC, applied specifically to the field of MSWI combustion status recognition. (2) High recognition accuracy: The constructed combustion status-recognition model exhibits superior accuracy in identifying various combustion statuses. (3) Online application validation: The application of the offline recognition model to online recognition systems demonstrates practical value and real-world applicability. (4) Real MSWI plant data: The research is based on actual MSWI plant flame data, offering important practical insights and guidance for implementation.

The study's limitations are apparent in two areas: (1) Incomplete representation: The considered combustion statuses might not encompass all the varied conditions observed

on site. Future work should involve supplementing these statuses based on expert insights to develop corresponding recognition models. (2) Qualitative analysis only: The current recognition model predominantly performs qualitative analysis of the flame's combustion status. There is a vital need to make quantitative analyses using flame data to assess factors like material layer thickness.

The flame combustion status online recognition system plays a pivotal role in boosting operational efficiency and reducing pollutant emissions within the MSWI process. This cutting-edge technology enables real-time monitoring of incineration flames, ensuring a consistently efficient and stable combustion process. Based on the software of the flame online-recognition system, precise control strategies can be employed to fine-tune combustion parameters, thus minimizing the release of harmful gases significantly and enhancing resource utilization efficiency. This intelligent-control approach contributes significantly to realizing the sustainability objectives of MSW management by combining incineration technology with environmental sustainability protection, steering the MSWI process toward a more eco-friendly direction.

**Author Contributions:** Methodology, J.T. and H.X.; Software, X.P.; Validation, T.W.; Formal analysis, H.X.; Investigation, T.W.; Writing—original draft, X.P.; Writing—review & editing, J.T.; Supervision, J.T. All authors have read and agreed to the published version of the manuscript.

**Funding:** This research received no external funding.

**Institutional Review Board Statement:** Not applicable.

**Informed Consent Statement:** Not applicable.

**Data Availability Statement:** Due to project restrictions, data will not be provided to the public.

**Conflicts of Interest:** The authors declare that they have no known competing financial interest or personal relationship that could have appeared to influence the work reported in this article.

## Nomenclature

| Symbols | Meaning |
| --- | --- |
| $I$ | Flame image |
| $y$ | Corresponding labels of flame-image dataset |
| $N$ | Number of flame-image datasets |
| $n$ | Index of flame image |
| $I^{\mathrm{Pre}}$ | Preprocessed image |
| $f_{Scale}$ | Image scaling operation |
| $f_{Gray}$ | Image grayscale processing |
| $j$ | Index of channel numbers in feature maps |
| $J$ | Number of feature map channels |
| $K$ | Convolutional kernel |
| $k$ | Elements in convolutional kernel |
| $b$ | Bias |
| $b$ | Bias element |
| $A$ | Output feature maps of convolutional and pooling layers |
| $a$ | Fully connected layer output feature map |
| $a$ | Output elements in feature maps |
| $mean(\cdot)$ | Matrix mean function |
| $*$ | Convolutional operation |
| $f_{\tanh}(\cdot)$ | Tanh activation function |
| $\hat{y}_n$ | Output of LeNet-5 |
| $e$ | The base of natural logarithms |
| $T$ | Number of categories |
| $down(\cdot)$ | Downsampling function |
| $Z$ | Net activation of convolutional layers |
| $z$ | Net activation of fully connected layers |
| $z$ | Elements in output feature maps |

| | |
|---|---|
| $\partial$ | Taking partial derivative |
| $\delta$ | Network middle-layer error |
| $C$ | Loss function |
| $\|\cdot\|_2$ | L2-norm |
| $\odot$ | Hadmard product |
| $\text{upsample}(\cdot)$ | Upsampling operation |
| $\text{ROT180}(\cdot)$ | Flip matrix 180 degrees |
| $U, V$ | Width and height of $\delta$ |
| $\boldsymbol{\theta}$ | General term for network parameters |
| $\alpha$ | Learning rate |
| $P$ | Total number of iterations for network training |
| $\nabla_p$ | Parameter gradient calculated during the $p$-th backpropagation |
| $S$ | Layer 1–4 output feature map of LeNet-5 |
| $s$ | Output feature flattening for each layer |
| $s^{\text{Fusion}}$ | Deep fusion features of flame images |
| Tree_Number | The number of decision trees in the CF layer forest |
| Mini_Samples | Minimum sample size of leaf nodes |
| $\hat{y}^{\text{Train}}$ | Offline recognition results |
| $f_{\text{DFC}}(\cdot)$ | DFC model |
| $\hat{y}$ | Online recognition results |
| TP | True positive example |
| FP | False positive example |
| TN | True negative example |
| FN | False negative example |
| $\dot{S}$ | Training set of RF |
| $G$ | Training subset of RF |
| $c$ | Index of RF training subset |
| $R^c$ | Number of features selected by the $c$-th training subset in the forest |
| $C$ | Count of bootstrap |
| $R_{\text{sel}}$ | Number of best segmentation feature |
| $s$ | The cut |
| $\text{Gini}(\cdot)$ | Index of Gini |
| $\theta_{\text{Forest}}$ | Threshold of the number of samples contained in leaf nodes |
| $y_{P_{\text{Left}}}$ | Label values corresponding to samples divided into left nodes in the training subset |
| $y_{P_{\text{Right}}}$ | Label values corresponding to samples divided into right nodes in the training subset |
| $\theta_{\text{Forest}}$ | Threshold of leaf node |
| $Q$ | Number of input feature space partition regions |
| $c_P$ | Class $c_P$ in dataset label $y$. |
| $p_{c_P}$ | The proportion of class $c_P$ to the total number of labels |
| $\Gamma^c(\cdot)$ | Classification tree model |
| $N_{G_q}$ | Number of training samples included in region $G_q$ |
| $\boldsymbol{y}^j_{N_{R_q}}$ | Label vectors corresponding to sample features in region $G_q$ |
| $\boldsymbol{p}^q_c$ | Prediction results of the final output of region $G_q$ |
| $\Lambda(\cdot)$ | Indicator function |
| $F_{\text{RF}}(\cdot)$ | RF model |
| $F_{\text{CRF}}(\cdot)$ | CRF model |

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
