# Peer review of "Online Combustion Status Recognition of Municipal Solid Waste Incineration Process Using DFC Based on Convolutional Multi-Layer Feature Fusion"

_sustainability, doi:10.3390/su152316473_

Round 1

Reviewer 1 Report

Comments and Suggestions for Authors

1. The dataset is important for the results of the model, and it is necessary to introduce the principle of dataset screening.

2. The study classifies the images of the grate combustion state into four categories, while the actual combustion process conditions are complex, it is necessary to introduce the basis for classification and the role of classification.

3. The manuscript provides a detailed description of the specific implementation of the algorithm. However, the steps need to be supplemented with an explanation of the basis of selection, such as why average pooling is used.

4. The online identification session needs to consider the time needed to apply the model.

Author Response

Point 1: The dataset is important for the results of the model, and it is necessary to introduce the principle of dataset screening.

Response 1: Thank you very much for your suggestion. We have added the principles of dataset screening on lines 584 to 589 as follows:

“Upon collection of flame videos from both the left and right grates onsite, the initial step involves the removal of fragments depicting unclear combustion statuses. Following this, the remaining video segments are classified according to the combustion status classification standard illustrated in Figure 2. These classified video segments are subsequently sampled at a consistent rate of 1 frame per minute utilizing a MATLAB program, resulting in the extraction of flame image frames. Consequently, the total count of typical combustion status images obtained from the left and right furnace bars is 3289 and 2685, respectively. For a detailed breakdown of each typical combustion status, please refer to Table 1.

Table 1. Flame Image Dataset.

Grate

Amount

Normal

Partial

Channelling

Smoldering

Size

Left

3289

655

1176

1044

414

720×576

Right

2685

564

1002

534

585

720×576

Point 2: The study classifies the images of the grate combustion state into four categories, while the actual combustion process conditions are complex, it is necessary to introduce the basis for classification and the role of classification.

Response 2:

According to your suggestion, we have added the criteria and functions of classification from line 211 to 226. The specific content is as follows:

“Before making the classification of combustion status, our preliminary investigation focused on abnormal combustion phenomena within biomass grate furnace combustion. Huang [15] defined layered combustion deviation status while studying diagnostic methods for the MSWI process, highlighting lateral and longitudinal deviations in the flame's spatial distribution. In the field of biomass grate furnace combustion, Duffy [35] and other researchers [36] identified a phenomenon termed "channeling." This occurs when the bed inside the combustion chamber is uneven or at the junction with the furnace's boundary wall. Channeling disrupts the uniformity of the secondary air blown in from beneath the grate, exacerbating bed irregularities. This article classifies MSWI flame images into four distinct combustion statuses: normal, deviation, channeling, and smoldering. This classification draws from the observed on-site flame combustion conditions in the studied MSWI process and the analysis of abnormal disturbance phenomena in grate furnace combustion, using knowledge from on-site experts and research scholars. Following the effective classification, corresponding adjustments to control strategies will be initiated based on the obtained results. Such initiatives, focused on artificial intelligence (AI) vision, will be a focal point for future research endeavors.”

Point 3: The manuscript provides a detailed description of the specific implementation of the algorithm. However, the steps need to be supplemented with an explanation of the basis of selection, such as why average pooling is used.

Response 3: According to your suggestion, we have added explanations for selecting average pooling and mean squared error loss. The specific content is as follows:

The explanation for choosing average pooling is as follows (lines 363 to 365):

“Compared to max pooling, average pooling helps to preserve the overall trend of the flame image and retain more background information, which is important for flame images.”

The explanation for choosing mean squared error loss is as follows (lines 423 to 425):

“When selecting the loss function, the mean squared error (MSE) is widely used due to its intuitive, easy-to-compute, and smooth characteristics.. So, the used loss function is the MSE loss, which is expressed as:

                           (14)”

Point 4: The online identification session needs to consider the time needed to apply the model.

Response 4: Your consideration is very necessary.

It has been added at the end of Section 4.3, which are as follows.

The hardware configuration used for building the model included an Intel(R) Core(TM) i9-11900K CPU, 32GB of RAM, and an NVIDIA GeForce RTX3060Ti GPU. The integrated development environment was MATLAB 2021b. The time required for the trained offline recognition model to analyze and recognize a flame image averaged approximately 0.174 seconds. During the online recognition process, flame images are sampled every minute to assess their combustion status. Given the relatively slow change in combustion status, this recognition speed effectively meets the requirements for real-time online recognition.

Reviewer 2 Report

Comments and Suggestions for Authors

This article proposes an online recognition method named flame video convolutional multilayer feature fusion deep forest classification (DFC). The authors reported that the proposed method achieves recognition accuracies of 93.80% and 95.08% for left and right grate furnace offline samples, respectively. This work is of significance. My main concerns are listed as follows.

ü  The novelty of this manuscript remains questionable, since the authors did not present the research gap in Online Combustion Status Recognition. Flame diagnosis is certainly of importance for the combustion industry. However, they have been extensively studied for several decades. Why did we further develop a new method for flame diagnosis? This is an essential issue for this topic. The authors should first present a review for this issue.

ü  Which toolbox did the authors employ to construct the model?

ü  Whether or not were the model parameters optimal for individual CNNs?

ü  Please give the definition of ‘PRECISION’ and ‘Recall’ in Table 4 and 5.

ü  Fig. 13 is not clear enough.

Author Response

Point 1: The novelty of this manuscript remains questionable, since the authors did not present the research gap in Online Combustion Status Recognition. Flame diagnosis is certainly of importance for the combustion industry. However, they have been extensively studied for several decades. Why did we further develop a new method for flame diagnosis? This is an essential issue for this topic. The authors should first present a review for this issue.

Response 1: Thank you very much for your suggestion.

We have added the following explanation regarding the necessity of writing this manuscript in lines 55 to 72:

“When it comes to recognizing combustion status through flame image analysis in the MSWI process, several studies exist, each focusing on different furnace types. Miyamoto et al. [9] conducted research on the "AI-VISION" system, integrating combustion image processing, neural networks for discerning combustion status, and online learning methods for optimizing neural networks. Their system manipulated operating values in fluidized bed incinerators. Zhou [10] developed a combustion status diagnosis model based on neural networks utilizing geometric features and grayscale information from flame images, validated through ten-fold cross-validation experiments. Guo et al. [11] presented a combustion status recognition method employing mixed data augmentation and a deep convolutional generative adversarial network (DCGAN) to obtain flame images under diverse conditions. Huang et al. [12] extracted key parameters like grayscale mean, flame area ratio, high-temperature ratio, and flame front to characterize and evaluate combustion status. Meanwhile, Zhang et al. [13] extracted 19 feature vectors encompassing color, shape, and texture of flame images, constructing an echo state network recognition model. These findings emphasize the necessity for further research and validation of combustion status identification methods tailored to different MSWI plants. In the field of combustion status recognition based on flame videos, researchers have proposed diverse solutions for similarly complex industrial processes.”

Point 2: Which toolbox did the authors employ to construct the model?

Response 2:

The tools we primarily use include MATLAB and its App Designer tool. Specifically, when building the offline recognition model, we write the program in MATLAB script files. When constructing the online recognition model, we design the front-end and back-end of software using both the graphical interface and code interface of App Designer.

Point 3: Whether or not were the model parameters optimal for individual CNNs?

Response 3: Yes, the recognition models for the burning state of the flame images from the left and right furnace grates are trained separately. The training results of the CNN models are shown in Figure 10. By training separate recognition models for each furnace grate, we ensure that the parameters of each CNN model are optimized individually.

Point 4: Please give the definition of ‘PRECISION’ and ‘Recall’ in Table 4 and 5.

Response 4: The definitions of 'precision' and 'recall' can be found in lines 604 and 605 respectively.

                                  (42)

                                  (43)

Point 5 Fig. 13 is not clear enough.

Response 5: Thank you very much for your suggestion.

We have made adjustments to the image clarity as per your recommendation.

Reviewer 3 Report

Comments and Suggestions for Authors

In this work, the authors conducted online combustion status recognition of municipal solid waste incineration process using DFC based on convolutional multi-layer feature fusion. Comments can be found below,

1.       As many kinds of previous methods including DFC are used in this area, I suggest the author give their advantage, disadvantage and application scenarios in the introduction part for a clear introduction.

2.       The novelty of your work should be clear and additionally highlighted, together with the objectives of your research, in the last paragraph of the Introduction.

3.       The figure of “Typical flame combustion state of MSWI process” in section 2.2 shows a bit confusing, more clear graphs and explanation would be better.

4.       There are no references in the section of methods, if any part in this section is not proposed by the authors, the original references are necessary.

5.       What are the uncertainties of the tests results?

6.       There are quite a few grammatical and editorial errors in the text.

Comments on the Quality of English Language

Extensive editing of English language required

Author Response

Point 1: As many kinds of previous methods including DFC are used in this area, I suggest the author give their advantage, disadvantage and application scenarios in the introduction part for a clear introduction.

Response 1: Thank you very much for your suggestion. We have added descriptions of the advantages, disadvantages, and application scenarios of Lenet5 and DFC in the article.

The advantages, disadvantages, and application scenarios of Lenet5 are described in lines 117 to 121 of the article, as follows:

“LeNet-5's capability to capture local image features based on local receptive fields, reduce network training parameters through shared weights, and maintain a simple network structure is noteworthy. Despite being an early convolutional neural network with shallow layers, LeNet-5 finds extensive use in image processing tasks like license plate recognition and face detection.”

The description of the advantages, disadvantages, and application scenarios of DFC has been added in lines 139 to 148 of the article. The specific text is as follows:

“Additionally, Nie et al. [32] proposed an online multi-view deep forest architecture for remote sensing image data. DFCs offer advantages over DNNs, such as fewer hyperparameters, interpretability, and automatic adjustment of model complexity [33]. Moreover, they perform well with smaller image data samples, effectively resolving challenges in constructing DNN recognition models. However, it's noteworthy that the multi-grained scan module of DFC can be time-consuming and inefficient in acquiring diverse scaled deep features. These studies collectively imply that DFC, combined with CNN-based deep feature extraction algorithms, can effectively tackle the limitations posed by limited flame image datasets in the MSWI process.”

Point 2: The novelty of your work should be clear and additionally highlighted, together with the objectives of your research, in the last paragraph of the Introduction.

Response 2: Thank you very much for your suggestion.

We have added the novelty of our work and research objectives in the last paragraph of the introduction (lines 163 to 173), as follows:

“The existing research highlights prevalent applications of online flame video recognition in areas like rotary kilns and electric magnesium melting furnaces. Surprisingly, there's a dearth of study regarding online flame video recognition in the MSWI field. Consequently, this article aims to explore an online recognition method tailored to the unique characteristics of flame videos in MSWI. The primary innovations of this method encompass: (1) Proposing a fusion technique that combines flame depth feature extraction and adaptive selection based on Lenet-5; (2) Integrating deep fusion features with the DFC algorithm to construct a combustion status recognition model specifically designed for the MSWI process; (3) Development of a practical online combustion status recognition platform based on flame video for MSWI. These advancements signify the potential practical value of this technology within the MSWI field.”

Point 3: The figure of “Typical flame combustion state of MSWI process” in section 2.2 shows a bit confusing, more clear graphs and explanation would be better.

Response 3: Thank you for your suggestion.

We have made modifications to Fig. 3 and added some explanatory text in lines 231 to 233 to make the flame image information more easily understandable. The specific changes are as follows:

  • Channeling burning (b) Smoldering       (c)  Partial burning    (d) Normal burning

Figure 3. Typical combustion status of MSWI process.

In Fig. 3, the red arrow's direction represents the flame's orientation, while the arrow's length corresponds to the flame's height. The blue line signifies the combustion line, while the red line outlines the outer flame's edge.”

Point 4: There are no references in the section of methods, if any part in this section is not proposed by the authors, the original references are necessary.

Response 4: Thank you for your suggestion.

We have added relevant references in the methods section, as follows:

Lines 317 to 318:

“Before inputting flame image dataset  into Lenet-5 [17], it needs to be preprocessed to meet the network input requirements.”

Lines 467 to 469:

“To enhance the model's performance, the DFC's multi-granularity scanning module [25] has been excluded, utilizing solely the CF module for constructing the combustion status recognition model.”

Point 5: What are the uncertainties of the tests results?

Response 5: In Fig. 14, the probability values for the current combustion state recognition results are obtained by the well-trained offline recognition model, which identifies the frames of the current sampled flame video. The probability value is calculated by averaging the classification results of each decision tree in the DFC model for the samples, followed by Soft-max normalization. The probability values indicate the relative confidence levels of the classification for each sample. Ultimately, the category with the highest probability value is displayed. The reason for using this probability representation is explained in lines 686 to 696.

“From Figure 14, it's evident that the software not only presents the current combustion status recognition results of the flame video but also assigns a probability value. This additional detail is reasonable due to the complexity of onsite combustion status, where distinct boundaries between the four categories of combustion status might not always be clear. In situations involving transitional or coupled phases of different combustion statuses, the probability representation mode is employed. This approach enables operators in practical MSWI plants to judge the confidence level of recognition results, providing a valuable reference for adjusting control strategies. It offers operators a clearer understanding of the degree of coupling within the current combustion status. Furthermore, this enhances the need for a dynamic recognition method based on contextual image correlation in future advancements.”

Point 6: There are quite a few grammatical and editorial errors in the text.

Response 6: In the new manuscript, we have carefully checked the manuscript for language, grammatical, and syntax errors and thoroughly corrected those errors.

Reviewer 4 Report

Comments and Suggestions for Authors

The manuscript “Online Combustion Status Recognition of Municipal Solid Waste Incineration Process Using DFC based on Convolutional Multi-Layer Feature Fusion” represents an online recognition method by using convolutional multi-layer feature fusion and DFC. Experimental results demonstrate that the proposed method achieves recognition accuracies of 93.80% and 95.08% for left and right grate furnace offline samples. When it is applied to an online flame video recognition platform, it fulfills the recognition demands. Although it is an interesting study, there are several aspects that must be reviewed before it can be accepted for publication.

Comment 1. Page 7 The network mainly consists of convolutional layer 1, poolinglayer 1, convolutional layer 2, pooling layer 2, fully connected layer 1, fully connected layer 2, and Softmax layer. ′ This section describes the composition and characteristics of the convolutional layer 1, poolinglayer 1, convolutional layer 2, pooling layer 2, fully connected layer 1 and fully connected layer 2. It is recommended to supplement the content of Softmax layer.

Comment 2. Page 19, There are some limitations in each stage for the method proposed in this article. Based on this limitation, it is suggested that the author supplement the proposed solutions and ideas for relevant issues.

Comment 3. Some earlier references should be replaced by the latest references.

Comment 4. There are some inappropriate English words or expressions in the manuscript. The authors should carefully polish the English of the whole manuscript.

Author Response

Point 1: Page 7 ′The network mainly consists of convolutional layer 1, poolinglayer 1, convolutional layer 2, pooling layer 2, fully connected layer 1, fully connected layer 2, and Softmax layer. ′ This section describes the composition and characteristics of the convolutional layer 1, pooling layer 1, convolutional layer 2, pooling layer 2, fully connected layer 1 and fully connected layer 2. It is recommended to supplement the content of Softmax layer.

Response 1: Thank you for pointing out our issue.

After careful examination, we found that the "Softmax layer" in line 303 is the "Output layer" described in line 382. Therefore, we changed the "Softmax layer" in Fig. 6 to "Output layer". The details are as follows::

Figure 6. Structure of Lenet5.

As shown in the Fig. 6, the network mainly consists of convolutional layer 1, pooling layer 1, convolutional layer 2, pooling layer 2, fully connected layer 1, fully connected layer 2, and output layer.

Point 2: Page 19, There are some limitations in each stage for the method proposed in this article. Based on this limitation, it is suggested that the author supplement the proposed solutions and ideas for relevant issues.

Response 2: We have supplemented the solutions and ideas for related issues in lines 659 to 682. The details are as follows:

“(1) In the data collection and analysis stage, the selection of typical video clips was meticulously done through expert labeling, excluding videos with severe combustion status coupling. Consequently, some unclear videos were not utilized in this study. In future research, these video clips might undergo denoising techniques before expert labeling is applied. Furthermore, the sampled video frames contain visual representations of various process data. For instance, Wang [38] utilized CCD radiation energy images to reconstruct the temperature distribution within the incineration system. Similarly, He et al. [39] measured flame radiation spectra to acquire temperature and emission rates of the burning flames, and Xie et al. [40] predicted calorific value by employing Yolov5 to identify waste types in images. Subsequent efforts will focus on integrating these images with the corresponding process data.

(2) In the offline modeling phase, the CNN-based feature extraction predominantly emphasizes local flame image features, potentially neglecting key global features essential for comprehensively observing complex combustion status within the MSWI process. In our prior investigations, we explored a combustion status recognition technique employing Vision Transformer-IDFC [41], leveraging the transformer's self-attention mechanism to extract significant global features from flame images, resulting in commendable recognition outcomes. Consequently, addressing the identification of complementary features and elimination of redundant ones becomes necessary, accomplished by employing feature selection methods that aim for maximal correlation and minimal redundancy. Additionally, optimizing classifier hyperparameters concurrently with those used in feature engineering can improve the generalization performance of the recognition model. To tackle this challenge, we aim to employ intelligent optimization algorithms inspired by biological intelligence, like genetic algorithms, differential evolution, and particle swarm optimization. Nevertheless, these approaches may introduce computational complexities with long running time. As a remedy, optimization using proxy models will be employed to address this new challenge.

(3) In the online recognition phase, we capture flame video frames at regular intervals and employ offline constructed recognition model for identification. The obtained recognition results are then fed back into the online recognition system, displaying them on the desktop. However, this process inherently employs a single-image identification method, lacking consideration for the temporal relationships and causal changes between image sequences over time. Flame videos, as a form of streaming data, encapsulate both spatial information within frames and temporal information between frames. Regrettably, the current recognition system neglects this temporal dimension. Techniques from other domains specializing in stream image mining and analysis, such as active learning with expert input [42], real-time video stream analytics [43], and streaming deep neural networks (DNN) [44], can be integrated. This enhancement would facilitate applications in the actual MSWI process, paving the way for intelligent control based on AI vision.

Point 3: Some earlier references should be replaced by the latest references.

Response 3: Thank you for your suggestion. We have replaced some early references as follows:

“Wang, H., Wang, H., Zhu, X., Song, L., Guo, Q., & Dong, F. (2021). Three-Dimensional Reconstruction of Dilute Bubbly Flow Field With Light-Field Images Based on Deep Learning Method. IEEE Sensors Journal, 21(12):13417-13429.”

“Huang, Z., Li, X., Du, H., Zou, W., Zhou, G., Mao, F., Fan, W., Xu, Y., Ni, C., Zhang, B., Chen, Q., Chen, J., & Hu, M. (2023). An Algorithm of Forest Age Estimation Based on the Forest Disturbance and Recovery Detection. IEEE Transactions on Geoscience and Remote Sensing, 61, 4409018.”

Point 4: There are some inappropriate English words or expressions in the manuscript. The authors should carefully polish the English of the whole manuscript.

Response 4: In the new manuscript, we have carefully checked the manuscript for language, grammatical, and syntax errors and thoroughly corrected those errors.

Reviewer 5 Report

Comments and Suggestions for Authors

Sustainability-2683102-report

Following the general guidelines the following questions have been addressed:

Novelty: Is the question original and well-defined? Do the results provide an advancement of the current knowledge? Yes, using the AI algorithms, the operation of waste incineration process can be made more effective and clean using the methods described in the work.

Scope: Does the work fit the journal scope*? No. The scope of Sustainability is “environmental, cultural, economic, and social sustainability of human beings”. The fact that waste incineration can be made more clean and is only a very basic fact. The article is about a detail on how to improve sustainability of a specific equipment. This is not sufficient to be in the scope.

Significance: Are the results interpreted appropriately? Are they significant? Are all conclusions justified and supported by the results? Are hypotheses carefully identified as such? Yes. The method is used correctly the conclusions are realistic and include description of the next steps.

Quality: Is the article written in an appropriate way? Are the data and analyses presented appropriately? Are the highest standards for presentation of the results used? Yes, but some detailed suggestions for minor corrections can be made. See below.

Scientific Soundness: Is the study correctly designed and technically sound? Are the analyses performed with the highest technical standards? Is the data robust enough to draw conclusions? Are the methods, tools, software, and reagents described with sufficient details to allow another researcher to reproduce the results? Is the raw data available and correct (where applicable)? Yes.

Interest to the Readers: Are the conclusions interesting for the readership of the journal? Will the paper attract a wide readership, or be of interest only to a limited number of people? (Please see the Aims and Scope of the journal.) The article is of interest to two groups of readers: those interested in using processing of camera observations as a tool for control and process improvement. Those interested in demonstration of the effectiveness of AI tools for this purpose.

Overall Merit: Is there an overall benefit to publishing this work? Does the work advance the current knowledge? Do the authors address an important long-standing question with smart experiments? Do the authors present a negative result of a valid scientific hypothesis? The results are an interesting step towards a final goal of control of process plant operation using AI.

English Level: Is the English language appropriate and understandable? Yes

Minor revisions are needed before publication in an appropriate journal.

1/

The explanations in lines 174-183 are not clear. “the combustion line in the dry section” must be defined. What is it. The descriptions “scattered”, “star”, “curve” are not clear. What does it mean?

2/

Due to limitations of the MS Word version used the equations at many places are not in line with the text. This should be corrected.

3/

In line 503-504 the groups of images considered is described. It is striking that a large majority of the figures is not normal and falls in the classes partial, channelling and smoldering. It has not been explained whether this overall division is created on purpose or corresponds to the actual operation of the incinerator in practice.

4/

The results in Table 4 are presented with a large number of digits. This suggests an accuracy that is unrealistic. Less digits should be displayed in accordance with their relevance.

Author Response

Point 1: The explanations in lines 174-183 are not clear. “the combustion line in the dry section” must be defined. What is it. The descriptions “scattered”, “star”, “curve” are not clear. What does it mean?

Response 1: Regarding the "combustion line in the dry section," it refers to the combustion line where the flame is located in the drying grate section. In order to help readers better understand this concept, we have added specific images of the drying grate, combustion grate, and ash grate, along with their corresponding flame images, from line 206 to line 213. It is shown as follows:

“Fig. 2 shows the correspondence relation between the distribution of the furnace grate inside the furnace and the flame image.

(a) Furnace grate image    (b) Flame image

Fig. 2. Correspondence between grate and flame image distribution

In Fig. 2. (a) shows the interior screen of the right-side furnace, which clearly shows the dry grate, combustion grates 1 and 2, burning grate, and steps between the grates. So, the flame-burning position can be determined by matching the furnace grate image.”

Then, from line 235 to line 237, we have included the definition of the combustion line as follows:

“In Fig. 3, the direction of the red arrow represents the direction of the flame, the length of the flame arrow represents the height of the flame, the blue line represents the combustion line, and the red line represents the edge line of the outer flame of the flame.”

Finally, "scattered," "star," and "curve" are used to describe the flame size during different combustion states such as smoldering, channeling burning, normal burning, and partial burning.

Point 2: Due to limitations of the MS Word version used the equations at many places are not in line with the text. This should be corrected.

Response 2: Thank you for your suggestions. We have carefully reviewed and revised all the equations and text in the entire document.

 Point 3: In line 503-504 the groups of images considered is described. It is striking that a large majority of the figures is not normal and falls in the classes partial, channelling and smoldering. It has not been explained whether this overall division is created on purpose or corresponds to the actual operation of the incinerator in practice.

Response 3: We collected flame video data from municipal solid waste incineration (MSWI) power plants from November 2020 to December 2022, totaling 132 hours and 30 minutes. Firstly, we selected flame video segments with clear images, no signal interference, distinct features, and relatively mild inter-state coupling, equivalent to clear and typical segments. Secondly, we need to clarify that the category of normal combustion is defined based on ideal combustion conditions, which have extremely strict requirements for the combustion process. However, due to the complex and variable composition of municipal solid waste and the co-incineration of sludge in China, it is difficult to ensure the appearance of flame images depicting normal combustion for extended periods during the actual combustion process. Therefore, the number of flame images depicting normal combustion state is less than those depicting abnormal combustion states.

Point 4: The results in Table 4 are presented with a large number of digits. This suggests an accuracy that is unrealistic. Less digits should be displayed in accordance with their relevance.

Response 4: Thank you very much for pointing out the error. Tables 4 and 5 are identical, with the only difference being the position of the grate. We have corrected the error in the header of Table 4 to match Table 5. It is shown as follows:

Table 4. Comparative experimental results of left grate.

Methods

Evaluation index

Accuracy

Precision

Recall

VGGnet

0.36893

0.09223

0.25

Mobilenet

0.81553

0.80217

0.75971

Densenet

0.83252

0.85054

0.78825

EfficientNet

0.55097

0.6452

0.60138

Regnet

0.7185

0.7124

0.7248

Lenet-5

0.8990

0.8986

0.8929

DFC

0.8832

0.8576

0.9022

Ours

0.9380

0.9182

0.9507

Table 5. Comparative experimental results of right grate.

Methods

Evaluation index

Accuracy

Precision

Recall

VGGnet

0.36418

0.09104

0.25

Mobilenet

0.77313

0.80396

0.75911

Densenet

0.87164

0.86668

0.88562

EfficientNet

0.77313

0.77245

0.77835

Regnet

0.8269

0.8211

0.8295

Lenet-5

0.9151

0.9122

0.9149

DFC

0.8942

0.8848

0.9001

Ours

0.9508

0.9456

0.9541

Round 2

Reviewer 2 Report

Comments and Suggestions for Authors

OK

Reviewer 3 Report

Comments and Suggestions for Authors

The authors have replied the former comments properly.